# Runoff component quantification and future streamflow projection in a large mountainous basin based on a multidata-constrained cryospheric-hydrological model

Mengjiao Zhang[1,2], Yi Nan[1,2], Fuqiang Tian[1,2]

[1]Department of Hydraulic Engineering, Tsinghua University, Beijing 100084, China

[2]National Key Laboratory of Hydrosphere Science and Hydraulic Engineering, Tsinghua University, Beijing 100084, China

*Correspondence to*: Yi Nan (ny1209@qq.com); Fuqiang Tian(tianfq@tsinghua.edu.cn)

**Abstract.** The Yarlung Tsangpo River (YTR) is one of the several major rivers originating from the Tibetan Plateau (TP). Large uncertainties existed in the studies related to streamflow variations in this basin, and the investigation is difficult due to the widely distributed snowpack, glaciers and permafrost and their complex effects on hydrological processes. In this study, we conducted a systematic analysis on the streamflow variations and runoff components in the YTR basin, using a physically-based hydrological model validated by streamflow and multiple datasets related to cryospheric processes. Main findings include (1) The contributions of snowmelt and glacier melt runoff to streamflow were limited, both for about 5~6% for the whole basin, which might be overestimated by previous studies. (2) The annual runoff would increase evidently in the future. The relative change of annual streamflow could exceed 90mm (~38%) at the outlet station in the far future compared to the historical period under the high emission scenario. (3) Adopting more observational data to calibrate the hydrological model played a critical role in reducing the uncertainty of hydrological simulation. The biases of snow and glacier simulation for data unconstrained led to a marked overestimation of contributions of snowmelt and glacier melt runoff to streamflow and further brought an underestimation of the increasing trends of annual runoff by approximately 5~10% in future projection. These results provide a relatively reliable reference of the streamflow change and the runoff components in both historical and future periods in the YTR basin, because more datasets were used to constrain the model uncertainty compared to previous studies.

## 1. Introduction

Change in streamflow and sediment in cold mountainous regions around the world has drawn great interest from researchers (Slosson et al., 2021; Zhang et al., 2023). The Tibetan Plateau (TP), as a typical cold mountainous region, widely known as the "Asian Water Tower", is the source region of many large rivers in Asia and plays a pivotal role in providing invaluable fresh water to its downstream countries. The hydrological changes in the TP region have drawn high attention for a long time and there have been numerous relevant researches on its hydrological process (Li et al., 2022; Zhang et al., 2022). However, further study is necessary to fully understand the streamflow conditions of the TP and there is still a lot of uncertainty in its runoff variations.

On the one hand, the special environmental conditions increase the complexity of hydrological processes in the TP region. Vast areas of snow, glaciers, permafrost and seasonally frozen ground distribute over the TP throughout the year and all cryospheric components can contribute to streamflow in various ways (Lan et al., 2014). Understanding their impact on hydrological processes is crucial for a confident prediction of runoff change under climate warming. Yet, this is a difficult task because the complex hydrological and cryospheric processes were typically insufficiently represented by hydrological models (Nan et al., 2022; Wang et al., 2024). On the other hand, marked atmospheric warming has changed the water balance of the TP and altered water resources in downstream countries (Yao et al., 2022). Remarkably, TP is one of the most significant regions responding to climate change and the effects of climate change on water availability differ substantially among basins (Immerzeel et al., 2010). Also, the continuous rising temperature leads to rapid retreat of perennial snow and glaciers, impacting runoff and regional water security as well (Chen et al., 2017).

The Yarlung Tsangpo River (YTR), also termed as Brahmaputra after it flows into India, is one of the several major rivers originating from the TP and the largest river system in the south TP. As a representative river basin of the TP, the dynamic interactions between cryosphere, hydrosphere and atmosphere are prominent in the YTR basin, in which the hydrological processes like snow and glacial melting are more vital compared to some other regions, and the hydrological processes are complicated and sensitive to climate changes with high uncertainty (Jiang et al., 2022; Xu et al., 2019).

Monitoring of hydrological stations is critical to investigate the changes in streamflow and is the prominent data source for related study. Observational evidence demonstrates substantial increases in both annual runoff and annual sediment fluxes in the headwaters of TP across the past six decades (Li et al., 2021). But further research on the composition and future changes of streamflow still relies on hydrological models for now. Distributed hydrological model is an essential tool for study on the hydrological process of basins while the difficulty is that the model parameters are physically insufficient with large uncertainty, due to the limited observation data to calibrate the model (Tian et al., 2020). There have already been many studies trying to simulate the hydrological processes more realistically, including considering the contributions of snow and glacier (Zhang et al., 2013; Chen et al., 2017), simulating seasonal permafrost (Wang et al., 2023), and developing tracer-aided hydrological models (Nan et al., 2022). However, the contribution of runoff components still has a significant uncertainty among different studies, and a consistent conclusion on this issue has yet to be reached. In specify, the estimated contribution of glacier melt to streamflow in the YTR basin ranges from 3.5% (Wang et al., 2021) to 29% (Boral and Sen, 2020). Besides, the reason for such divergence remains unclear, and the influence of runoff component estimation on future streamflow projection has not been investigated adequately. A reliable reference value for runoff components obtained by robust modeling method is crucial for water resource management.

In this study, we conducted a systematic analysis on the streamflow change in the YTR basin based on observation streamflow data and various datasets related to cryospheric processes. We focused on the streamflow change during the historical period, the contribution of multiple runoff components, and the trend in the future period. We conducted different calibration variants to evaluate the value of different datasets on the model performance and the consequent impacts on the

runoff component partitioning and future projection results. We structured the paper into the following sections. Section 1 formulates the background of this study. Section 2 briefly introduces the YBR basin, followed by the used materials and methods. The main results are presented in Section 3. A brief discussion including a comparison with previous studies are in Section 4, followed by the conclusions in Section 5.

## 2. Materials and methodology

### 2.1 The Yarlung Tsangpo River

Located on the north of the Himalaya Mountains in the southern TP, the YTR originates from the Gyima Yangzoin glacier at the northern foot of the Himalayas and then travels through China, Bhutan, and India before emptying into the Bay of Bengal in the Indian Ocean. The length of the main stream is over 2000 km and there are four streamflow gauging stations distributed along it, including Lazi, Nugesha, Yangcun, and Nuxia station from upstream to downstream (Tian et al., 2020, red triangle in Fig. 1). The Nuxia station near the border of the TP is selected as the basin outlet of the study area, with a total drainage area of approximately $2 \times 10^5$ km² (Fig. 1). The average elevation of the YTR basin is about 4850m a.s.l. (above sea level), with an extent of 1890–6840m.

The mean temperature of the basin is relatively low (~ –3.1℃, 1979–2018) due to the high altitude, while the precipitation is mostly driven by the South Asian monsoon, with an average annual precipitation of about 475mm (1979–2018). Large amounts of moisture from the Indian Ocean entering the plateau water cycle through precipitation can significantly supplement its water resources (Zhou et al., 2019), with an obvious wet season from June to September, which accounts for 60–70% of the total annual rainfall (Xu et al., 2019). Moreover, the changes of the precipitation and runoff demonstrate strong consistency in the exoreic TP rivers, including the YTR (Tian et al., 2023). The average snow cover area is 16.8%, and glaciers cover ~2.1% of the basin (He et al., 2021), resulting in a considerable contribution of meltwater to runoff.

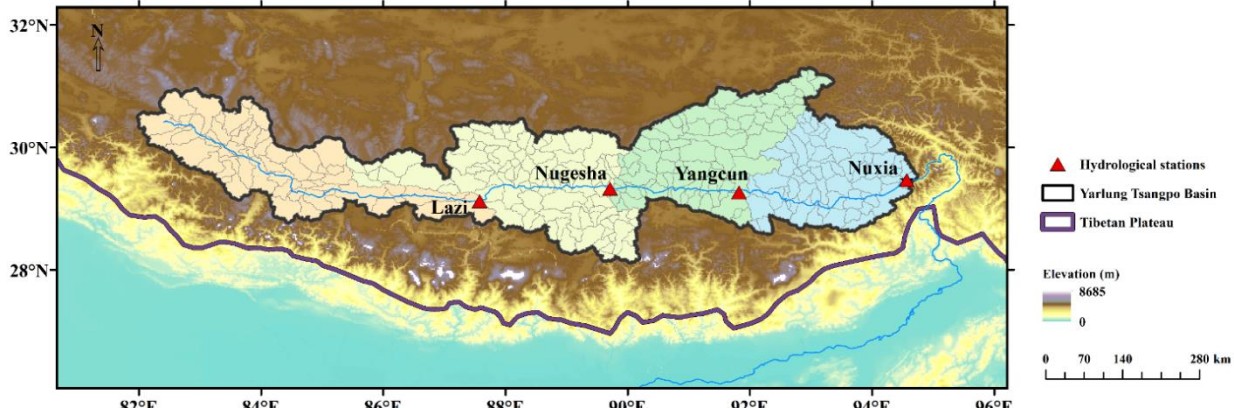

**Figure 1** Study area and locations of the hydrological stations.

## 2.2 Data

### 2.2.1 Hydrological station data

Extensive streamflow measurements were collected at four hydrological stations for variation analysis and hydrological model evaluation. The monthly/annual observations during 1960–2020 were obtained for trend testing, and daily data covering the model simulating period were obtained for model calibration. It should be noted that due to data

confidentiality requirements, the measured discharge in the results part were not presented directly by normalization or
hiding the vertical coordinates.
**Table 1** Basic information of hydrological stations used in the study area.

| Station | Longitude (°E) | Latitude (°N) | Altitude (m) | Drainage area (km²) | Period of observational streamflow | |
|---|---|---|---|---|---|---|
| | | | | | Daily | Monthly / Annual |
| Lazi | 87.576 | 29.121 | 4003 | 52516 | 1980–2020 | 1960–2020 |
| Nugesha | 89.712 | 29.325 | 3850 | 113758 | 1960–2020 | 1960–2020 |
| Yangcun | 91.822 | 29.266 | 3627 | 164518 | 1960–2020 | 1960–2020 |
| Nuxia | 94.567 | 29.467 | 2955 | 206019 | 1960–2020 | 1960–2020 |


## 2.2.2 Data for model driving and calibration

Daily meteorological inputs mainly include precipitation, temperature, and potential evapotranspiration (PET).
Precipitation data of the YTR basin were collected from the 0.1° grid China Meteorological Forcing Dataset (CMFD, Yang
et al., 2019) while temperature and potential evapotranspiration were obtained from the 1.0° grid reanalysis dataset
ERA5_Land for the historical calibration and validation periods. Underlying surface inputs consist of topography, glacier,
vegetation coverages and soil parameters. Elevation was derived from a digital elevation model (DEM) with a spatial
resolution of 30 m from the Geospatial Data Cloud (https://www.gscloud.cn). The second glacier inventory data set of
China (Liu, 2012) was used to denote the glacier coverage. Vegetation coverages were extracted from the MODIS satellite
products of 8-day leaf area index (LAI) dataset MOD15A2H (Myneni et al., 2015) and monthly normalized difference
vegetation index (NDVI) dataset MOD13A3 (Didan, 2015). Soil types and properties were collected from Global high-
resolution data set of soil hydraulic and thermal parameters (Dai et al., 2019). For future hydrological simulations, data
from 10 CMIP6 (Coupled Model Intercomparison Project Phase 6, https://esgfnode.llnl.gov/search/cmip6/) GCMs was
used as climate inputs, with more detailed introduction in 2.2.3.
For the calibration in the historical periods, in addition to the observational daily streamflow during 1980–2018 at the
four stations mentioned in 2.2.1, datasets of snow and glacier were adopted to evaluate the hydrological model. The snow
depth (SD) dataset for TP (Yan et al., 2021)), the Tibetan Plateau Snow Cover Extent product (TPSCE, Chen et al., 2018)
and the Glacier mass balance data (Hugonnet et al., 2021) were used to calibrate SWE (snow water equivalent), SCA (snow
cover area) and GMB (glacier mass balance) respectively. More details of the datasets above can be found in Table 2. Here,
the SD measurements were transferred to SWE for calibration using the following Eq. (1) (Chen et al., 2017):
$$SWE = \frac{\rho_{snow} \times SD}{\rho_{water}} = \frac{0.1966 \times SD^{0.9063}}{\rho_{water}} \tag{1}$$

where $\rho_{snow}$ is the snow density, $SD$ is the PMV-based (PMV, i.e. Passive Microwave) snow depth of snowpack, and
$\rho_{water}$ is the density of liquid water. The coefficients were estimated by in situ data.
**Table 2** Data from global and regional datasets used for hydrological models in this study.

| Datasets as inputs of the hydrological model | | | | |
|---|---|---|---|---|
| Dataset | Source/Name | Temporal resolution /Period | Description/Notes | Reference and/or Website for download |
| Precipitation | CMFD (China Meteorological Forcing Dataset) | Daily, 1979–2018 | 0.1° grid, its accuracy for China is better than that of the internationally available reanalysis data | Yang et al. (2019) |
| Temperature | ERA5_Land | Daily, 1950–2020 | 1.0° grid, a reanalysis dataset providing a consistent view of the evolution of land variables over several decades at an enhanced | https://cds.climate.copernicus.eu/cdsapp#!/dataset/ |
| PET (potential evapotranspiration) | | | | |

| | | | resolution compared to ERA5 | |
|---|---|---|---|---|
| Topography | SRTM DEM | – | 30m spatial resolution | https://www.gscloud.cn/ |
| NDVI (normalized difference vegetation index) | MOD13A2 | Monthly, 2000–2020 | 0.5 arc degree grid, derived from the Advanced Very High Resolution Radiometer (AVHRR) sensors | Didan et al. (2015) |
| LAI (Leaf Area Index) | MOD15A2H | 8-day, 2000–2020 | 0.05° grid, derived from the Advanced Very High Resolution Radiometer (AVHRR) sensors | Myneni et al. (2015) |
| Soil | Global high-resolution data set of soil hydraulic and thermal parameters | – | Optimal soil water retention parameters obtained from ensemble pedotransfer functions | Dai et al. (2019), http://globalchange.bnu.edu.cn/research |
| Glacier distribution | SCGI (The second glacier inventory data set of China) | 2006–2011 | Clear and concise overview and scientific assessment of the glaciers in China | Liu et al. (2012) |
| Climate (Precipitation and Temperature) | CMIP6 GCMs | Daily, ~2100 | More details in Table 3 | https://esgfnode.llnl.gov/search/cmip6/ |
| **Datasets for calibration of the hydrological model** | | | | |
| Observational streamflow | Relevant hydrology Bureau | Daily, ~2020 | More details in Table 1 | – |
| SD (snow depth) | A daily, 0.05° Snow depth dataset for Tibetan Plateau (2000–2018) | Daily, 2000/9/1–2018/8/31 | 0.05° grid, based on the snow cover probability in the Tibetan Plateau | Yan et al. (2021) |
| SCA (snow cover area) | TPSCE (Long-term TP daily 5-km cloud-free snow cover extent record) | Daily, 1981/8/1–2014/12/31 | 5-km cloud-free snow cover extent record derived from AVHRR surface reflectance CDR | Chen et al. (2018) |
| GMB (glacier mass balance) | Glacier mass balance data | Annual, 2000–2019 | standardized observations on changes in mass, volume, area and length of glaciers over time | Hugonnet et al. (2021) |


### 2.2.3 Bias-corrected GCMs data

The general circulation models (GCMs) are commonly used to simulate the earth's climate change and project the future climate change under a suite of different possible emission scenarios. Coupled Model Intercomparison Project Phase 6 (CMIP6) is the latest available CMIP simulations, which is improved comparing to the previous phase. Nevertheless, the CMIP6 GCMs still have diverse deviations at the regional scale. Taking the TP region for instance, most models underestimate the observed trends in mean and extreme temperature and precipitation (Cui et al., 2021).

We evaluated the performance of 22 CMIP6 GCM products and finally chose 10 GCMs to conduct this study, based on the stability of these data in the hydrological model and the rationality of simulation results. The basic information of these 10 GCMs is shown in Table 3. The CMIP6 data during 1960–2100 (divided into historical and future periods by 2014) were interpolated from various spatial resolutions into the same degree (0.1° grid) through a bilinear interpolation scheme. The biases in the GCMs data were further corrected against the reanalysis meteorological data (CMFD for precipitation and ERA5_Land for temperature, using 1979–2009 as the reference period for correction, and 2010–2018 for validation) based on a multiplicative bias-correction approach (MBCn algorithm, Alex J. Cannon, 2018; Cui et al., 2023). The average precipitation and temperature of the corrected GCMs are presented in Fig. 2. After bias correction, the overestimation on precipitation and temperature by GCMs was corrected, but uncertainties still existed in different GCMs. In specify, different GCMs produced a 14.3 mm/yr and 0.27°C difference in mean annual precipitation and temperature for historical period. For the future period, these differences increased to 68.32/62.78/102.43 mm/yr for precipitation and 1.01/1.01/1.66°C for temperature under SSP1-2.6, SSP2-4.5, and SSP5-8.5 scenarios, respectively. When driving the future model, the future PET data was calculated with CMIP6 temperature data and the historical temperature-PET correlation, as shown in Eq. (2) below (Cui et al., 2023), and other input data was kept same as the historical period.

$$PET = [1 + \alpha_0(T - \overline{T}_0] \cdot \overline{PET}_0 \tag{2}$$

where $\overline{T}_0$ and $\overline{PET}_0$ are the daily mean temperature (in °C) and potential evapotranspiration (in mm day$^{-1}$) for the
calendar month during the period 1979-2009 (provided by the ERA5_Land), respectively; $T$ is the daily temperature from
the CMIP6 model output (in °C); $\alpha_0$ is determined for each calendar month by regressing the ERA5_Land-based PET to
daily temperature over each grid.

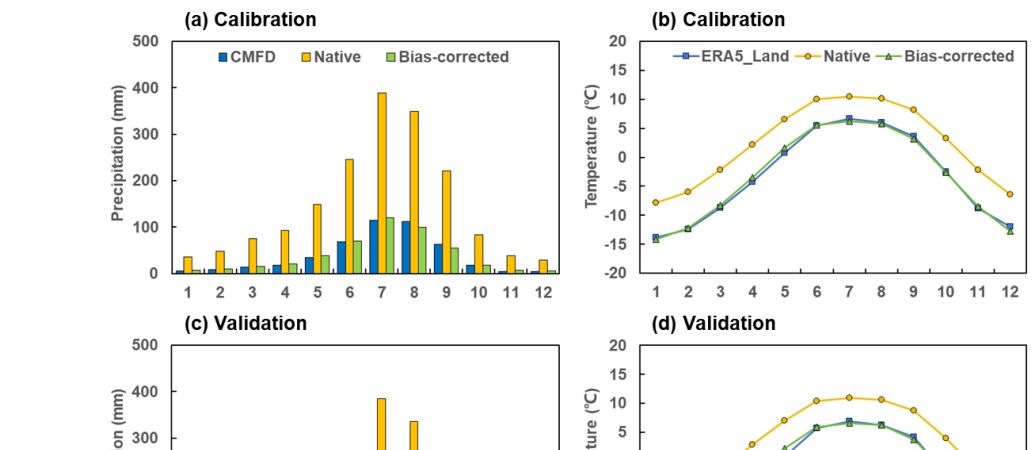

**Figure 2** Seasonal cycles of precipitation ((a) and (c)) and temperature ((b) and (d)) calculated from the historical data (CMFD/ERA5_Land), the ensemble
mean of 10 native and bias-corrected CMIP6 data during the calibration (1979–2009) and validation (2010–2018) period.
**Table 3** Basic information of ten CMIP6 GCMs used in this study.

| No. | Name | Nation | Resolution (Lon×Lat) | Period |
|---|---|---|---|---|
| 1 | ACCESS-ESM1-5 | Australia | 1.875°×1.2143° | 1950–2100 |
| 2 | BCC-CSM2-MR | China | 1.125°×1.125° | 1950–2100 |
| 3 | CNRM-CM6-1 | France | 1.40625°×1.40625° | 1950–2100 |
| 4 | GFDL-ESM4 | U.S. | 1.25°×1° | 1950–2100 |
| 5 | INM-CM5-0 | Russia | 2°×1.5° | 1950–2100 |
| 6 | MIROC6 | Japan | 1.40625°×1.40625° | 1960–2100 |
| 7 | MPI-ESM1-2-HR | Germany | 0.9375°×0.9375° | 1960–2100 |
| 8 | MPI-ESM1-2-LR | Germany | 1.875°×1.875° | 1950–2100 |
| 9 | MRI-ESM2-0 | Japan | 1.125°×1.125° | 1950–2100 |
| 10 | NESM3 | China | 1.875°×1.875° | 1950–2100 |

**2.3 Hydrological model**
A spatially-distributed physically-based hydrological model, the Tsinghua Representative Elementary Watershed
(THREW) model (Tian et al., 2006) was adopted to simulate streamflow of the YTR basin. This model uses the
representative elementary watershed (REW) method for spatial discretization of catchments (Reggiani et al., 1999) and the
YTR basin was divided into 276 REWs based on DEM data, as shown in Fig. 1. Areal averages of the gridded estimates
of meteorological variables, vegetation cover, soil property, and CMIP6 data were calculated in each REW to drive the
model.

For application in cold mountainous regions, the THREW model is incorporated with modules characterizing
cryospheric hydrological processes including snowpack dynamics and glacier evolution, and has been successfully applied
in several basins across China and the world (Xu et al., 2019; Tian et al., 2020; Nan et al., 2022; Cui et al., 2023). In the
THREW model, the degree-day method was used to simulate snow and glacier melting, assuming that snow and glaciers
melt at different rates (i.e., different degree-day factors), and relevant parameters including temperature thresholds were
calibrated. The snow water equivalent in each REW was updated based on the snowfall and snowmelt, and the snow cover
area was then determined by the snow cover depletion curve. To represent the change in meteorological factors along the
altitudinal profile of glaciers, each REW was further divided into several elevation bands to simulate the evolution of
glaciers. For each glacier simulation unit, processes including the snow accumulation and snowmelt over a glacier, the
turnover of snow to ice, and the ice melt were considered. The mass balance of each glacier simulation unit equaled the
difference between snowfall on glacier and the total meltwater. A detailed description of the snow and glacier modules and
the related equations could be found in Cui et al. (2023).

Here a modification was made upon the simulation of snowpack accumulation and melting processes on the basis of
the model in Cui et al. (2023). The snow sublimation was newly taken into account, similar to Han et al. (2019). In specify,
the amount of snowfall entering the runoff-generation process was deducted by a certain proportion of sublimation and two
additional parameters were introduced for this simulation. The details of the calibrated parameters of the THREW model
in this study could be found in Supplementary Table 1.

There are two definitions to quantify the contributions of runoff components to streamflow in the THREW model.
One was based on the individual water sources in the total water input triggering runoff processes, including rainfall,
snowmelt, and glacier melt and another was based on pathways of runoff-generation pathway, resulting in surface and
subsurface runoff (baseflow) (Nan et al. 2022). Here we focused on the first definition and calculated the contributions of
different water sources (rainfall, snowmelt, and glacier melt) to the total runoff. More precisely, the terms snowmelt and
glacier melt refer to meltwater from snow and glaciers, which enters the catchment and drives runoff generation processes
without having undergone evaporation, and the total discharge was equal to the sum of these three components minus
evaporation, thereby achieving the water balance in the THREW model.

## 2.4 Model calibration

Considering the time period of multiple datasets (the most applicative precipitation data over the YTR basin to build
the model covered 1979-2018), the simulation period was selected as 1980-2018, and was divided into two periods by 2009
(i.e. 1980–2009 for calibration and 2010–2018 for validation). Automatic calibration was implemented by the pySOT
(Python Surrogate Optimization Toolbox) algorithm to obtain the multiple-optimal objective (Eriksson et al. 2019). The
Nash-Sutcliffe efficiency coefficient (NSE) and the logarithmic Nash-Sutcliffe efficiency coefficient (lnNSE) were used
together to optimize the simulation of discharge, which can assess the simulations of both high flow and baseflow processes.
The root mean square error (RMSE) was used for the evaluation of SWE, SCA and GMB simulation. More details about
these metrics are presented in Table 4. The datasets for calibration in Table 2 was used separately with the corresponding
model outputs to calculate these evaluation indicators in model calibration.

To assess the effect of various datasets on calibration and their impact on simulation results, in addition to the scenario
taking all the elements (discharge, SWE, SCA, GMB) into consideration, we deleted different elements from the calibration
objectives to form different comparative variants. Thus, there were four variants for comparison: (1) D, calibration solely
using discharge, (2) DG, calibration using discharge and GMB, (3) DS, calibration using discharge, SWE and SCA, (4)
DSG, calibration using discharge, SWE, SCA and GMB. A plainer description of calibration variant designation was shown

in Table 5. For these variants, the model is calibrated for the whole basin, i.e., the discharge of basin outlet (Nuxia station) and the basin-scale average values of other elements (SWE, SCA, GMB) were compared between simulations and observations to evaluate the model. Correspondingly, the value of parameter was assumed to be universal for all the REWs of the basin.

Furthermore, an additional variant was added on the basis of variant "DSG", referred to as "ALL". It also considered all elements, but the discharge data at upstream stations were used for calibration to better consider the spatial heterogeneity within the basin. In the "ALL" variant, the model used four different sets of parameters for the four sub-regions divided by four hydrological stations.

In each calibration variant, the pySOT program was repeated for 100 times to obtain adequate parameter samples. A final parameter set was selected from the 100 calibrated sets manually based on the overall performance on multiple objectives. The adopted parameters of the THREW model in the YTR basin by all calibration variants are provided in Supplementary Table 2.

**Table 4** The calibration elements and the metrics used to evaluate the model performance in this study.

| Element | Timescale | Unit | Metrics | Formula | Range | Ideal value |
|---|---|---|---|---|---|---|
| Discharge | Daily | m³/s | NSE (Nash Sutcliffe coefficient) | $NSE = 1 - \dfrac{\sum_{i=1}^{n} (Q_{O,i} - Q_{S,i})^2}{\sum_{i=1}^{n} (Q_{O,i} - \overline{Q_O})^2}$ | $(-\infty, 1)$ | 1 |
| | | | lnNSE (logarithmic Nash Sutcliffe efficiency coefficient) | $lnNSE = 1 - \dfrac{\sum_{i=1}^{n} (lnQ_{O,i} - lnQ_{S,i})^2}{\sum_{i=1}^{n} (lnQ_{O,i} - \overline{lnQ_O})^2}$ | $(-\infty, 1)$ | 1 |
| SWE | | cm | RMSE (Root mean square error) | $RMSE = \sqrt{\dfrac{\sum_{i=1}^{n} (A_{O,i} - A_{S,i})^2}{n}}$ ("A" can be replaced by SWE, SCA or GMB) | $(0, +\infty)$ | 0 |
| SCA | | – | | | | |
| GMB | Annual | m/a | | | | |
| Note: n is the total number of observations, subscripts of "o" and "s" refer to observed and simulated variables, respectively. | | | | | | |

**Table 5** Five calibration variants of the THREW model in this study.

| No. | Objective of calibration | Abbreviation | Notes |
|---|---|---|---|
| 1 | Discharge | D | Only discharge was considered |
| 2 | Discharge + GMB | DG | Snow elements not considered |
| 3 | Discharge + SWE + SCA | DS | Glacier element not considered |
| 4 | Discharge + SWE + SCA + GMB | DSG | All elements were considered (Variant "ALL" used 4 stations, while the others used Nuxia station only) |
| 5 | | ALL | |

## 2.5 Analysis on the streamflow change

### 2.5.1 Historical trend

The past 6 decades (1960-2020) was selected to analyze historical streamflow changes based on the start time of the measurement at hydrological stations, and to analyze the trend and change-point of streamflow, the Pettitt test and linear regression methods were adopted with the monthly/annual runoff observations at the four hydrological stations (Zhang et al., 2024). Pettitt test is a non-parametric approach to the change-point problem (Pettitt, 1979), which can be used for mutation analysis of hydrological sequences to test the abrupt change points. And after obtaining the abrupt change point of the runoff in the historical period (1960–2020), if the periods divided by it is still long (＞20a), the test will be conducted again to obtain the abrupt change points relative to the primary abrupt change point. Linear regression method is commonly used to analyze the long-term evolution characteristics of hydrological sequences, reflecting the overall trend and then

providing guidance for water resource utilization. Here the linear regression method was used to calculate the rate of change
and the t-test method was used to determine the significance, quantitatively reflecting the variation trend of runoff over
time.
2.5.2 Future projection

As mentioned in section 2.2.3, 10 CMIP6 GCMs were used in this study and bias correction has been conducted upon
the GCMs based on observation data. Although the bias correction process modified the mean values of precipitation and
temperature, their variation characteristics in the future were mostly preserved, exhibiting significantly rising precipitation
and temperature in the future (Supplementary Fig. 1). Then the observation-constrained THREW model in the YTR basin
was driven by the bias-corrected CMIP6 data under the historical period (1960–2014) and the future period (2015–2100)
under three Shared Socioeconomic Pathways (SSPs) scenarios, i.e., SSP 1-2.6 (SSP126), SSP 2-4.5 (SSP245) and SSP 5-
8.5 (SSP585). The results simulated by models of different calibration variants, and under different future SSP scenarios
were both compared in this study. In the meantime, considering the time period for model calibration and GCMs' bias
correction, the results during 1980–2009 was used as the baseline of historical simulation and two periods (2020–2049 as
Near future, 2070–2099 as Far future) were selected as representatives for the future simulation. The relative changes of
streamflow in these two future periods compared to the historical period and the contributions of different runoff
components to discharge in these representative time periods were particularly calculated to evaluate the future changes.
Different time periods were adopted in different analyses. In summary, the past 6 decades (1960-2020) was selected
as the time period of historical streamflow trend analysis, based on the available time period of measurement streamflow
data. The simulation period was selected as 1980-2018 because the most applicable precipitation input dataset over the
YTR basin (CMFD dataset) only covered this period, and was further divided into two periods by 2009 for model
calibration (1980-2009) and validation (2010-2018). The future projection analysis adopted 1960-2014 and 2015-2100 as
historical and future periods, because the CMIP6 GCMs divided the historical and future periods by 2014, while the
historical period here had several years of overlap with the simulation period. Consequently, three periods were selected to
represent the baseline historical period (1980-2009), near future (2020-2049) and far future (2070-2099).

**3. Results**
**3.1 Streamflow change characteristic during the historical period**

As shown in Table 6, the annual runoff at four stations in the YTR basin did not exhibit a significant trend over the
past six decades. The annual runoff of the three upper stations (Lazi, Nugesha and Yangcun) showed a deceasing trend
while that of outlet station (Nuxia) exhibited an increasing trend, but all these trends were insignificant. Figure 3 presents
the annual runoff process divided by abrupt change years at the four stations. The change-point of annual runoff was
different among four stations, but 1998 was a common turning year when an abrupt runoff change occurred at three of the
stations.
Figure 4 shows the average monthly runoff at four stations. The runoff was mostly contributed by summer (June to
August) and autumn (September to November) runoff, accounting for ~50% and ~30% of the annual runoff, respectively
(Table 6). As for the spatial variation, the measured runoff at different stations appeared to be consistent overall, showing
similar intra-annual distribution of monthly runoff, but the changing rates of annual and seasonal runoff were different
among stations. The summer and winter runoff at the four stations all displayed a decreasing and increasing trend,
respectively, while the changes of autumn runoff were all consistent with the annual runoff. The spring runoff at the upper
stations (Lazi and Nugesha) displayed significant changes.

**Table 6** Abrupt change points and trend testing results of annual and seasonal streamflow in the historical period (1960–2020) at the four hydrological stations of the YTR basin.

| Station | Abrupt change points of annual streamflow | | Variation trends of annual and seasonal streamflow (mm/a) [a] | | | | | Contributions of seasonal streamflow to the annual streamflow (%) | | | |
|---|---|---|---|---|---|---|---|---|---|---|---|
| | Primary | Secondary | Annual | Spring | Summer | Autumn | Winter | Spring | Summer | Autumn | Winter |
| Lazi | 1965 | 2017 | −0.16 | +0.23* | −0.58 | −0.27 | +0.01 | 12.0 | 48.8 | 30.2 | 9.0 |
| Nugesha | 1972 | 1998 | −0.16 | −0.16* | −0.37 | −0.22 | +0.05 | 9.1 | 50.7 | 31.8 | 8.4 |
| Yangcun | 1998 | 1981、2005 | −0.09 | −0.02 | −0.05 | −0.30 | +0.11 | 8.0 | 52.0 | 32.2 | 7.8 |
| Nuxia | 1998 | 1981、2005 | +0.02 | +0.10 | −0.20 | +0.15 | +0.14 | 9.4 | 53.1 | 30.5 | 7.0 |

a: The annotation "*" indicates a significant change (at the 0.05 level of significance).

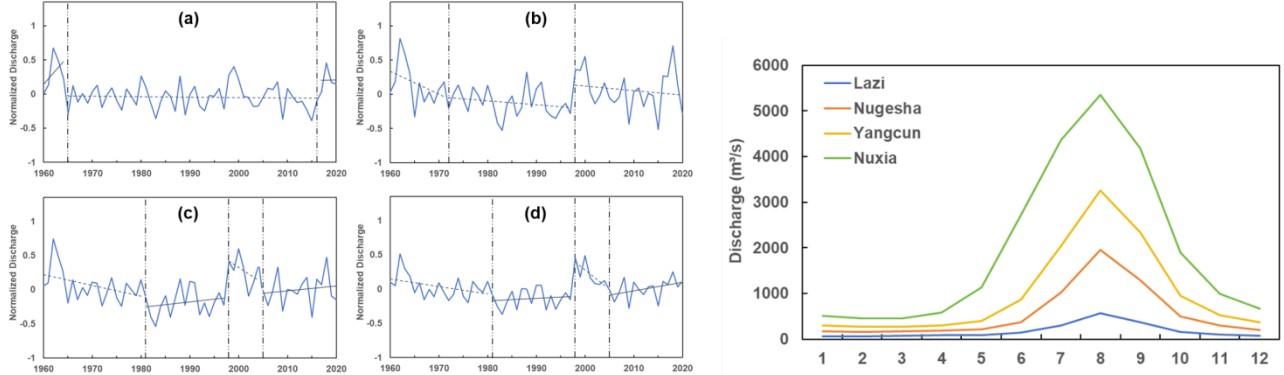

**Figure 3** (Left) Annual runoff process divided by abrupt change years at the 4 stations of the YTR basin. (a)–(d) for Lazi, Nugesha, Yangcun and Nuxia, respectively.

**Figure 4** (Right) Average monthly runoff during 1960–2020 at the 4 stations of the YTR basin.

## 3.2 Model performance obtained by different calibration variant

Figure 5 shows the observed and simulated discharges at the Nuxia station for calibration and validation periods by various calibration variants. The THREW model performed well in discharge simulation under these variants and almost all of their NSE and lnNSE values during the calibration and validation periods were beyond 0.8, with some of them exceeding 0.9. But with regard to the simulation of other elements, different variants performed variously. The performances of SWE, SCA and GMB simulations and the specific evaluation metrics are shown in Fig. 6 and Table 7.

Seasonal and interannual variations in SWE, SCA and GMB were reproduced well by calibration variant "DSG", indicated by the low values of $RMSE_{SWE}$, $RMSE_{SCA}$, and $RMSE_{GMB}$. Due to the uncertainty of the observed Snow Depth product and the relatively simplified calculation process of SWE, the variations of SWE were not simulated as well as other elements, but the average simulated SWE was close to the average observation, indicating that the amount of snowpack was reproduced well. In comparison, variant "DG" significantly overestimated the SWE, as indicated by the high $RMSE_{SWE}$, while an obvious overestimation of GMB simulation occurred in the variant "DS", with a high value of $RMSE_{GMB}$. The variant "D" performed the worst overall, along with the most significant overestimation of SWE, obvious bias of GMB and high values of $RMSE_{SWE}$ and $RMSE_{GMB}$. For the calibration of snow, SWE played a more pronounced constraint role, while SCA's constrain was easier to be satisfied. The values of $RMSE_{SCA}$ in these four variants were all relatively low (~0.10), but the simulated SCA processes of variant "DG" and "D" were higher than observation, while the peaks were a bit underestimated by other two variants (Fig. 6).

To summarize, variations of all elements (discharge, SWE, SCA and GMB) were reproduced well by calibration variant "DSG", effectively utilizing all observed data. Comparatively, variant "DG" and "DS" performed poorly in the simulation of snow and glacier process respectively, whereas the single-objective variant "D" presented poor performances

in simulation of all the elements except for the discharge. Thus, among these four different variants, arguably the variant "DSG" with the most objectives in calibration could achieve the comprehensively best result.

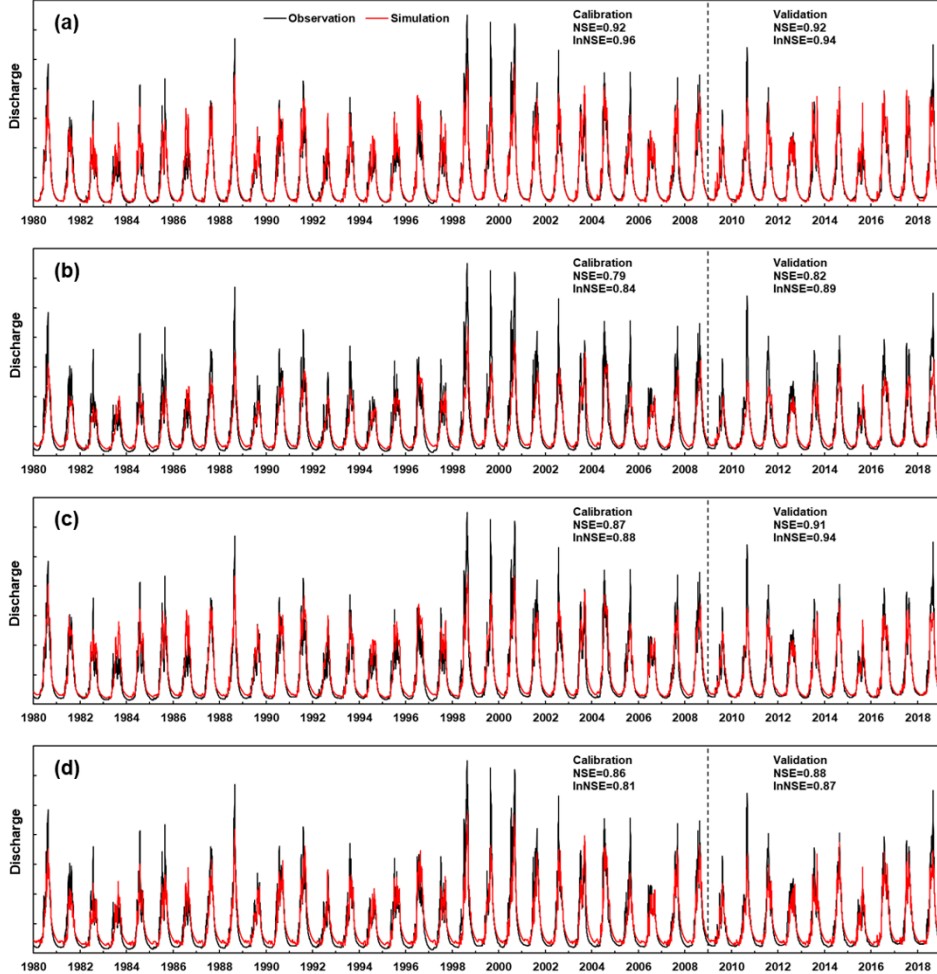

**Figure 5** Annual discharge processes of observation and simulation at the Nuxia station during 1980–2018 by various calibration variants. (a)–(d) for calibration variant "D", "DG", "DS", "DSG", respectively. The discharge data are hidden due to the data confidentiality regulations, and same for figures 7, 9 and 10.

**Table 7** Calibrated and validated results at Nuxia station by various calibration variants.

| Element (unit) | Metrics | Period | Calibration variant | | | |
|---|---|---|---|---|---|---|
| | | | D | DG | DS | DSG |
| Discharge | NSE | 1980–2009 | 0.92 | 0.79 | 0.87 | 0.86 |
| | | 2010–2018 | 0.92 | 0.82 | 0.91 | 0.88 |
| | | 1980–2018 | 0.92 | 0.80 | 0.88 | 0.87 |
| | lnNSE | 1980–2009 | 0.96 | 0.84 | 0.88 | 0.81 |
| | | 2010–2018 | 0.94 | 0.89 | 0.94 | 0.87 |
| | | 1980–2018 | 0.95 | 0.85 | 0.89 | 0.82 |
| SWE (cm) | RMSE | 2000–2009 | 1.79 | 1.20 | 0.19 | 0.24 |
| | | 2010–2018 | 2.30 | 1.61 | 0.27 | 0.33 |
| | | 2000–2018 | 2.05 | 1.41 | 0.23 | 0.28 |
| SCA | RMSE | 1981–2009 | 0.07 | 0.06 | 0.12 | 0.10 |
| | | 2010–2014 | 0.13 | 0.08 | 0.10 | 0.08 |
| | | 1981–2014 | 0.08 | 0.07 | 0.11 | 0.10 |

| GMB (m/a) | RMSE | 2000–2009 | 0.14 | 0.10 | 1.20 | 0.12 |
|---|---|---|---|---|---|---|
| | | 2010–2018 | 0.28 | 0.20 | 1.07 | 0.17 |
| | | 2000–2018 | 0.22 | 0.15 | 1.14 | 0.14 |

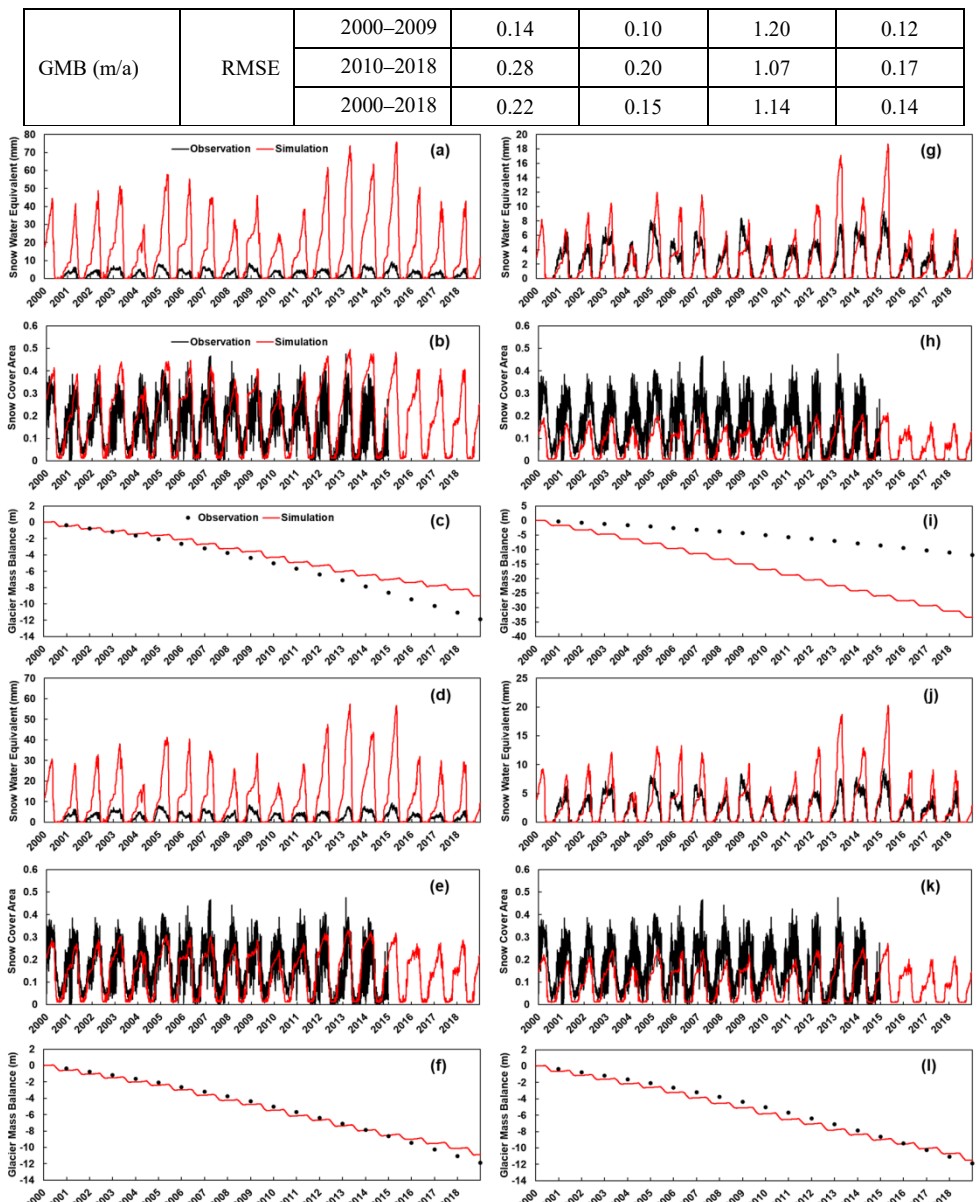

**Figure 6** Annual processes of SWE, SCA and GMB of observation and simulation in the whole YTR basin (Nuxia station) during 2000–2018 by various calibration variants. (a–c), (d–f), (g–i) and (j–l) for calibration variant "D", "DG", "DS", "DSG", respectively.

Then we further focused on the simulations of upstream stations and the calibration variant "ALL" was set as a supplement. The simulation results at all hydrological stations in the YTR basin by calibration variant "DSG" and "ALL" are shown in Table 8. Although the two variants achieved similar performance at the outlet station (Nuxia), both well reproducing the processes of discharge, SWE, SCA and GMB, there were significant differences in the results at the upstream stations. The variant "ALL" obviously performed better in the simulation of upstream stations, with high values of NSE and lnNSE (NSE＞0.8 and lnNSE＞0.7 at Yangcun and Nugesha stations, NSE and lnNSE ＞0.6 at Lazi station) and low values of $RMSE_{SWE}$, $RMSE_{SCA}$, and $RMSE_{GMB}$ during the calibration and validation periods, while variant "DSG" had significant deviations, especially for the most upstream Lazi station. Therefore, variant "ALL" was considered to have further improvements compared to variant "DSG", which could better simulate the hydrological processes in different regions of the basin. Figure 7 and 8 present the observed and simulated discharges and other calibration elements at all stations of the YTR basin under the variant "ALL". The simulated discharge process of all stations coincided with the observed process on the whole, and for the processes of SWE, SCA and GMB in different regions, they were also close to

the observed processes overall. For comparison, the simulations at upstream stations under the variant "DSG" are shown in Supplementary Figures 2 and 3. The variant "DSG" produced an abnormal fluctuation in discharge during baseflow period at upstream stations, resulting in extremely low values of lnNSE. The snow and glacier simulations were also worse than the variant "ALL", showing larger RMSEs for SWE, SCA and GMB simulations.

**Table 8** Calibrated and validated results at all hydrological stations in the YTR basin by calibration variant "DSG" and "ALL".

| Element (unit) | Calibration/Validation /the entire study period | | DSG | | | | ALL | | | |
|---|---|---|---|---|---|---|---|---|---|---|
| | | | Nuxia | Yangcun | Nugesha | Lazi | Nuxia | Yangcun | Nugesha | Lazi |
| Discharge | NSE | 1980–2009 | 0.86 | 0.80 | 0.66 | −0.31 | 0.85 | 0.88 | 0.82 | 0.66 |
| | | 2010–2018 | 0.88 | 0.80 | 0.72 | −0.24 | 0.84 | 0.83 | 0.75 | 0.67 |
| | | 1980–2018 | 0.87 | 0.80 | 0.68 | −0.29 | 0.85 | 0.86 | 0.80 | 0.66 |
| | lnNSE | 1980–2009 | 0.81 | 0.51 | 0.19 | −0.48 | 0.92 | 0.84 | 0.74 | 0.72 |
| | | 2010–2018 | 0.87 | 0.58 | 0.31 | −0.58 | 0.93 | 0.83 | 0.74 | 0.69 |
| | | 1980–2018 | 0.82 | 0.52 | 0.22 | −0.50 | 0.92 | 0.83 | 0.74 | 0.72 |
| SWE (cm) | RMSE | 2000–2009 | 0.24 | 0.29 | 0.38 | 0.73 | 0.21 | 0.25 | 0.34 | 0.68 |
| | | 2010–2018 | 0.33 | 0.42 | 0.56 | 1.07 | 0.29 | 0.37 | 0.50 | 1.02 |
| | | 2000–2018 | 0.28 | 0.36 | 0.48 | 0.91 | 0.25 | 0.31 | 0.42 | 0.86 |
| SCA | RMSE | 1981–2009 | 0.10 | 0.07 | 0.06 | 0.11 | 0.11 | 0.08 | 0.05 | 0.09 |
| | | 2010–2014 | 0.08 | 0.07 | 0.08 | 0.14 | 0.09 | 0.07 | 0.06 | 0.11 |
| | | 1981–2014 | 0.10 | 0.07 | 0.06 | 0.12 | 0.11 | 0.07 | 0.05 | 0.10 |
| GMB (m/a) | RMSE | 2000–2009 | 0.12 | 0.16 | 0.12 | 0.07 | 0.08 | 0.07 | 0.08 | 0.15 |
| | | 2010–2018 | 0.17 | 0.25 | 0.26 | 0.21 | 0.21 | 0.16 | 0.17 | 0.20 |
| | | 2000–2018 | 0.14 | 0.21 | 0.20 | 0.15 | 0.15 | 0.12 | 0.13 | 0.18 |

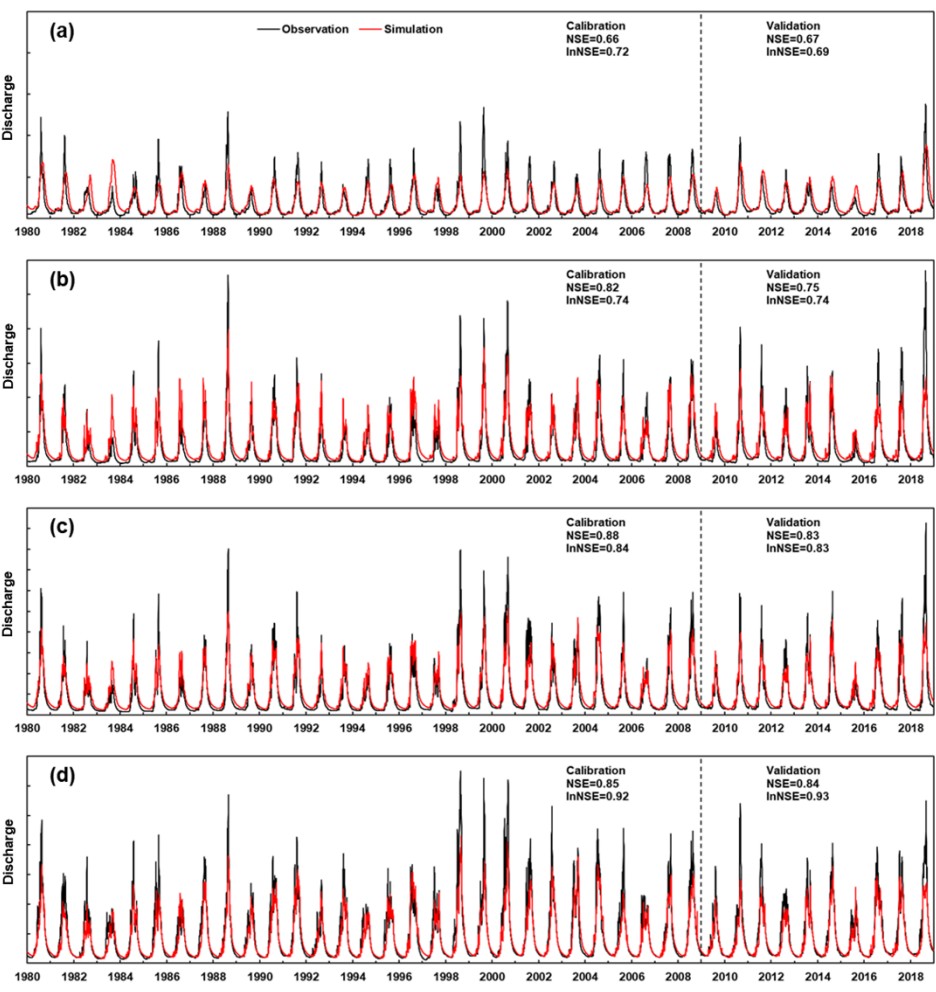

**Figure 7** Annual discharge processes of observation and simulation at four stations in the YTR basin during 1980–2018 by calibration variant "ALL".
(a)–(d) for Lazi, Nugesha, Yangcun, Nuxia station, respectively.

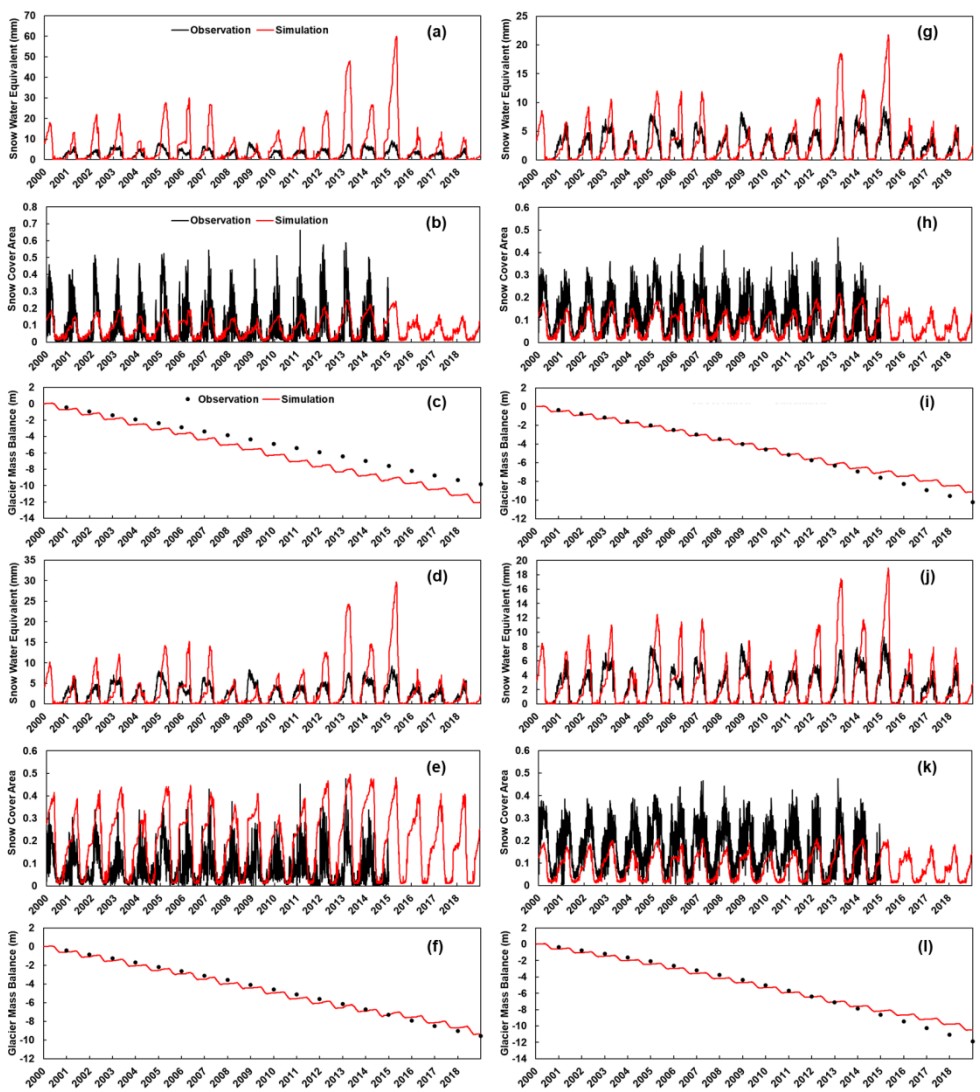

**Figure 8** Annual processes of SWE, SCA and GMB of observation and simulation in various regions of the YTR basin during 2000–2018 by calibration variant "ALL". (a–c), (d–f), (g–i) and (j–l) for the region to Lazi, Nugesha, Yangcun, Nuxia station, respectively.

## 3.3 Contributions of each runoff component to streamflow

Table 9 shows the contributions of different runoff components to streamflow during the simulation period (1980–2018) at Nuxia station estimated by various calibration variants. Although the discharge variation was well reproduced by all variants in the calibration and validation periods, the contributions of runoff components were quite different among different variants. In all calibration variants, rainfall was the dominant water source with contribution higher than 70%. The contribution of glacier melt was estimated lower than 10%, while the contribution of snowmelt varied significantly among different variants. For the calibration variant "DSG", the mean contributions of rainfall, snowmelt, and glacier melt to annual streamflow in the YTR basin were around 88.8%, 4.9%, and 6.3%, respectively. In variant "DG", the contribution of glacier melt to streamflow was 6.3%, same as that of the "DSG" variant, but the contribution of snowmelt was much higher (16.7%). Conversely in variant "DS", the contribution of snowmelt to streamflow was 4.5%, close to that of the variant "DSG", yet the contribution of glacier melt was higher (9.7%). Regarding the variant "D", the contributions of runoff component were similar to variant "DG", but the contribution of snowmelt was even higher, close to 20%. The differences above in contributions of runoff components were basically consistent with the model performance on the simulation of each element.

Comparing the variant "DSG" and 'ALL", the contributions of runoff components to streamflow at the Nuxia station
obtained by the two variants were similar, with snowmelt and glacier melt together accounting for 11~12%. However, as
for upstream stations, the contributions of meltwater runoff in the upstream stations under variant "DSG" were quite small
(＜10% at Yangcun and Nugesha stations, and ＜20% at Lazi station), while the result obtained by variant "ALL" was a
bit different. Snowmelt and glacier melt runoff accounted for a larger proportion in upstream stations. The contributions of
snowmelt and glacier melt runoff during 1980–2018 were 7.5%, 5.1% at Yangcun station, 8.9%, 5.3% at Nugesha station,
and 23.9%, 11.6% at Lazi station, respectively. The contributions of snowmelt and glacier melt runoff in different regions
would vary due to factors like the difference in snow and glacier coverage within the region, and the spatial variation of
degree-day factors (Zhang et al., 2006). Owing to the calibration results at upstream stations, the runoff composition results
at different stations under the variant "ALL" were believed to be more reasonable.
**Table 9** Contributions of different runoff components to discharge during 1980–2018 at the Nuxia station by various calibration variants and at upper
stations by calibration variant "DSG" and "ALL".

| Component (%) | Calibration variant / station | | | | | | | | | | |
|---|---|---|---|---|---|---|---|---|---|---|---|
| | D | DG | DS | DSG | | | | ALL | | | |
| | Nuxia | | | Nuxia | Yangcun | Nugesha | Lazi | Nuxia | Yangcun | Nugesha | Lazi |
| Rainfall | 74.4 | 77.0 | 85.8 | 88.8 | 90.9 | 90.3 | 82.7 | 87.8 | 87.4 | 85.8 | 64.5 |
| Snowmelt | 19.6 | 16.7 | 4.5 | 4.9 | 5.1 | 5.8 | 10.3 | 6.0 | 7.5 | 8.9 | 23.9 |
| Glacier | 6.0 | 6.3 | 9.7 | 6.3 | 4.0 | 3.9 | 7.0 | 6.2 | 5.1 | 5.3 | 11.6 |


## 3.4 Projection of future streamflow

Figure 9 shows the average annual discharge simulated with 10 CMIP6 GCMs during 1960–2100 at the Nuxia station
by the model calibrated by four variants. The streamflow projections generated by the 10 GCMs exhibited substantial
variation, ranging from 60% to 160% of the average streamflow, as indicated by the uncertainty bars in Figure 9. To address
this variability, we used the average of the 10 GCMs to represent the ensemble projection result. In spite of the deviations
among GCMs and the different parameters obtained by different calibration variants, the annual mean streamflow in the
YTR basin was projected to increase consistently in the future. The runoff increased insignificantly under SSP126 and
SSP245 scenarios, while the increasing trend under SSP585 scenario was visible, with the P value <0.01 in all time periods
under SSP585 scenario. Figure 9 also shows the relative changes of annual discharge under three SSP scenarios in the near
and far future period. Here we can find that under some SSP scenarios (mainly SSP245 and SSP585), there could also be
a slight decrease in total runoff in the near future, which was compatible with the results in the previous study (Cui et al.,
2023). But in the far future, the total runoff showed a notable increase under three SSP scenarios by all calibration variants.
For instance, under the calibration variant "DSG", the relative change of annual streamflow depth at the Nuxia station was
6.0mm (2.2%) / -3.0mm (-1.1%) / -13.1mm (-4.8%) under SSP126/245/585 scenario respectively in the near future period
(2020-2049) compared to the historical period (1960–2009), and was 16.2mm (6.0%) / 31.4mm (11.6%) / 90.9mm (33.6%)
for the same condition in the future period (2070–2099).
Table 10 provides the specific average variation trends during different periods simulated with 10 CMIP6 GCMs.
Under different variants, the increasing trend of streamflow under SSP585 scenario at Nuxia Station were all projected to
exceed 1.7mm/a during the future period (2015–2100), especially the far future period (all ＞2.3mm/a). But under SSP126
scenario, the annual total streamflow showed a downward trend in the far future period and under SSP245 scenario, the
variation trends of streamflow in the far future period were low (most ＜0.1mm/a). Moreover, the future streamflow of all
the upstream stations also presented an increasing trend (Fig. 10), but their increasing trends were not so significant as the

outlet station (Nuxia). Under the variant "ALL", the variation trend of streamflow at Nuxia station was about 1.92mm/a during 2015–2100 under SSP585 scenario, while the trends of streamflow at Yangcun, Nugesha and Lazi station were 1.47, 0.99 and 0.50mm/a, respectively. Similar to the Nuxia station, the total runoff of the upstream station exhibited relatively small changes in the near future period, while showed significant changes in the far future period. Compared to the historical period, the relative change of annual streamflow depth in the far future period was 26.6mm (102.8%), 50.3mm (57.7%), 76.2mm (51.0%) and 94.6mm (39.9%), respectively at Lazi, Nugesha, Yangcun and Nuxia station under SSP585 scenario.

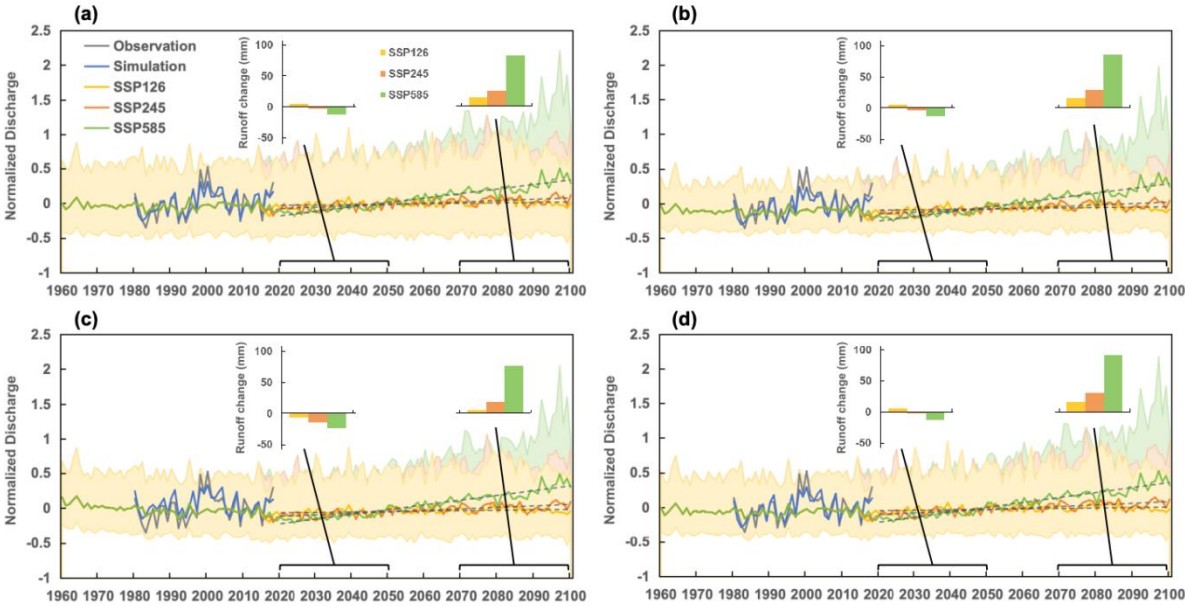

**Figure 9** Average annual discharge simulated with 10 CMIP6 GCMs during 1960–2100 at the Nuxia station by the calibrated model by four variants (the grey line is the observed value, the blue line is the simulated value in calibration, and the shaded area indicates the deviations of 10 GCMs data). (a)–(d) for calibration variant "D", "DG", "DS", "DSG", respectively.

**Table 10** Average variation trends during different periods at the Nuxia station by various calibration variants and at upper stations by calibration variant "ALL" simulated with 10 CMIP6 GCMs.

| Variation trend (mm/a) | | Calibration variant / station | | | | | | | |
|---|---|---|---|---|---|---|---|---|---|
| | | D | DG | DS | DSG | ALL | | | |
| | | Nuxia | | | | Nuxia | Yangcun | Nugesha | Lazi |
| 1960–2014 | | 0.01 | 0.05 | −0.69 | 0.03 | 0.07 | 0.12 | 0.05 | 0.03 |
| 2015–2100 | SSP126 | 0.25 | 0.25 | 0.24 | 0.27 | 0.27 | 0.24 | 0.18 | 0.08 |
| | SSP245 | 0.57 | 0.62 | 0.62 | 0.68 | 0.69 | 0.56 | 0.40 | 0.23 |
| | SSP585 | 1.73 | 1.81 | 1.82 | 1.92 | 1.92 | 1.47 | 0.99 | 0.50 |
| 1980–2009 (Historical) | | 0.47 | 0.55 | 0.05 | 0.52 | 0.56 | 0.55 | 0.30 | 0.22 |
| 2020–2049 (N-Fu) | SSP126 | 0.54 | 0.60 | 0.40 | 0.59 | 0.58 | 0.49 | 0.37 | 0.26 |
| | SSP245 | 0.45 | 0.52 | 0.40 | 0.57 | 0.57 | 0.45 | 0.34 | 0.25 |
| | SSP585 | 1.21 | 1.25 | 1.09 | 1.30 | 1.25 | 1.04 | 0.74 | 0.38 |
| 2070–2099 (F-Fu) | SSP126 | −0.21 | −0.33 | −0.21 | −0.28 | −0.30 | −0.14 | −0.07 | −0.12 |
| | SSP245 | 0.03 | −0.02 | 0.15 | 0.11 | 0.08 | 0.05 | 0.03 | 0.04 |
| | SSP585 | 2.40 | 2.37 | 2.63 | 2.60 | 2.50 | 1.94 | 1.34 | 0.59 |

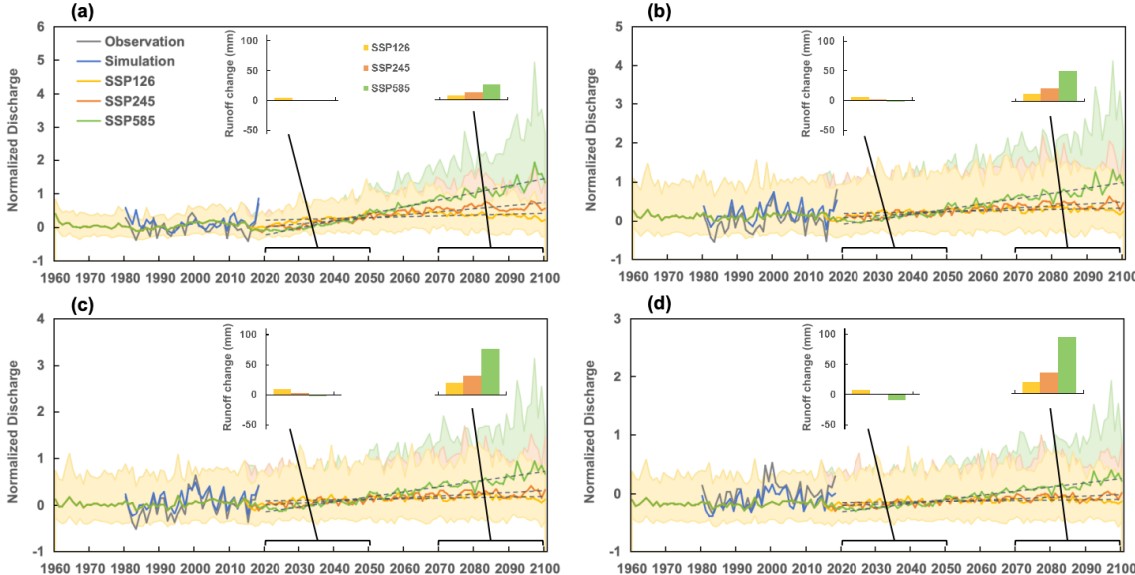

**Figure 10** Average annual discharge simulated with 10 CMIP6 GCMs during 1960–2100 at four stations in the YTR basin by the calibrated model by calibration variant "ALL" (the grey line is the observed value, the blue line is the simulated value in calibration, and the shaded area indicates the deviations of 10 GCMs data). (a)–(d) for Lazi, Nugesha, Yangcun, Nuxia station, respectively.

Despite the similar future trend of total streamflow, the changes of its components were different among variants, as shown in Fig. 11. With the rising precipitation and temperature, the contributions of both snowmelt and glacier melt would decrease in the future. The decreasing trend of snowmelt/glacier melt runoff was more rapid in the variants estimating higher contributions of the corresponding runoff component. The amounts and contribution proportions of snowmelt and glacier melt runoff exhibited a significant decreasing trend, regardless of the calibration variants and SSP scenarios. The decreasing snowmelt runoff was due to the reduced snowfall caused by climate warming, while the reduced glacier melt runoff indicated that the effect of shrinking glacier areas was more dominant than the acceleration of glacier melting caused by global warming. For instance, in the calibration variant "DSG", the glacier area in the YTR basin by the end of 2100 was only about 37%, 33% and 25% of that in 2010s under three SSP scenarios, respectively.

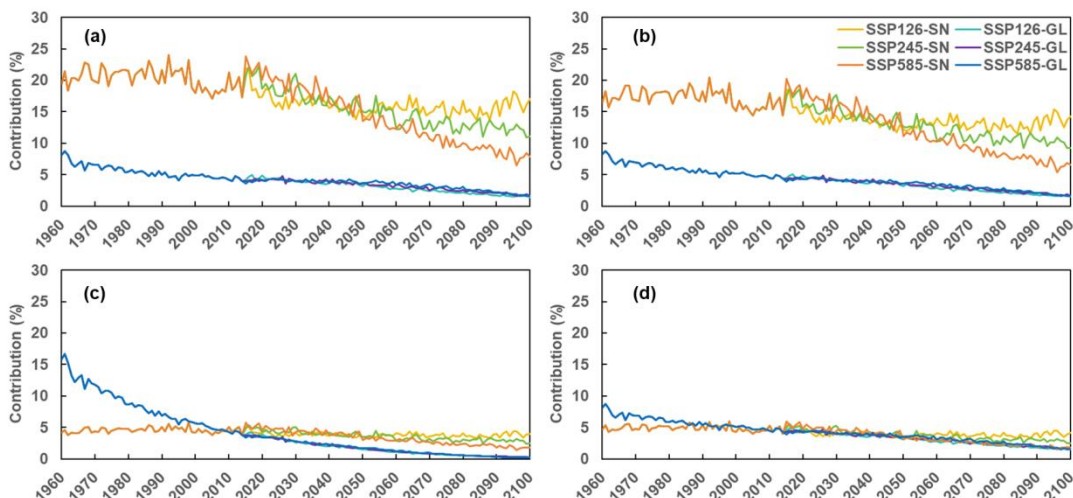

**Figure 11** Average annual snowmelt runoff and glacier melt runoff simulated with 10 CMIP6 GCMs during 1960–2100 at the Nuxia station by the calibrated model by four calibration variants (The abbreviation SN and GL represent snowmelt and glacier melt runoff, respectively). (a)–(d) for calibration variant "D", "DG", "DS", "DSG", respectively.

More visible results of the changes of various runoff compositions can be seen in Fig. 12 and 13, which shows the
relative changes of annual discharge and different runoff components under three SSP scenarios in the near future (2020–
2049) and far future period (2070–2099), respectively, compared to the historical period (1980–2009) at the Nuxia station
estimated by four calibration variants. The reduction in snowmelt runoff was most notable under SSP585 scenario in the
far future due to the most significant increase in temperature, while the reduction of glacier melt runoff did not differ that
significantly under different SSP scenarios. The contribution of meltwater in variant "DSG" was relatively small, so the
decrease in meltwater runoff due to the rising temperature played a less significant role, and the increase in total runoff in
the future was more significant compared to other calibration variants, which was also reflected by the more significant
variation trends of streamflow in variant "DSG" (Table 10). The most significant decreasing streamflow was estimated by
the "DS" calibration variant that estimated the highest contribution of glacier melt runoff among variants, which seemed
counterintuitive. This is because of the most significant shrinkage of glacier coverage area caused by the fast glacier melting
rate compared to other variants. In specify, the glacier area in the YTR basin by the end of 2049 simulated by the "DS"
variant was only about 40% of that in 2010s under SSP245 scenario, while this proportion was approximately 68% for the
"DSG" variant.

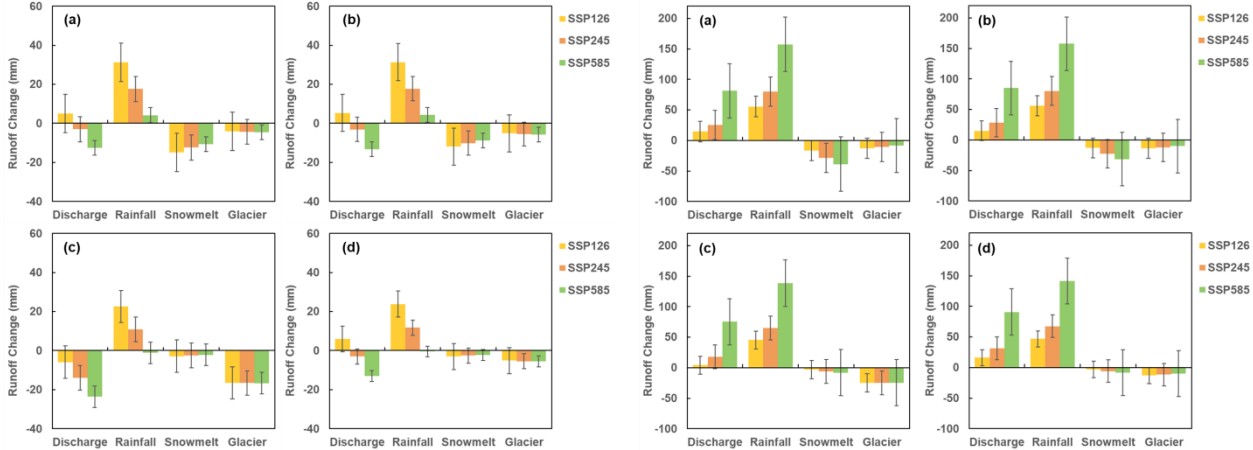


**Figure 12** (Left) Relative changes of annual discharge and different runoff components under three SSP scenarios in the Near future period (2020–2049)
compared to the historical period (1980–2009) at the Nuxia station estimated by four calibration variants (error bars represent one standard deviation).
(a)–(d) for calibration variant "D", "DG", "DS", "DSG", respectively
**Figure 13** (Right) Relative changes of annual discharge and different runoff components under three SSP scenarios in the Far future period (2070–2099)
compared to the historical period (1980–2009) at the Nuxia station estimated by four calibration variants (error bars represent one standard deviation).
(a)–(d) for calibration variant "D", "DG", "DS", "DSG", respectively.

Figure 14 presents the average contributions of different runoff components to discharge in different periods under
SSP585 scenario at the Nuxia station estimated by four calibration variants. The contributions of runoff components in the
historical period estimated by the model driven by the bias-corrected CMIP6 data was similar to that driven by original
input dataset (CMFD and ERA5), illustrated by the "Sim" and "His" columns. Under the most extreme scenario (i.e.
SSP585), the sum contribution of snowmelt and glacier melt runoff could decrease from 10% to less than 5% in calibration
variant "DSG" and "DS" (Fig. 14 (a) and (c)), and from over 20% to less than 10% in variant "DG" and "D" (Fig. 14 (b)
and (d)), in which the contribution of glacier melt runoff would be only about 1~2% under SSP585 scenario in the far
future. Because of the high contribution of rainfall runoff, the increasing precipitation was the determining factor causing
the rising future runoff in the YTR basin, and the rainfall runoff would play a more dominant role in the total runoff in the
near and far future periods compared to the historical period.

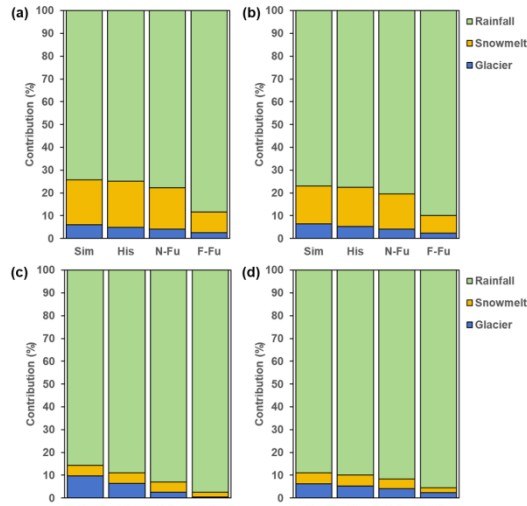

**Figure 14** Contributions of different runoff components to discharge in the calibration period (i.e. 1980–2018, represented by "Sim"), and the historical period (1980–2009), near future period (2020–2049), and far future period (2070–2099) under SSP585 scenario (represented by "His", "N-Fu" and "F-Fu", respectively) at the Nuxia station estimated by four calibration variants. (a)–(d) for calibration variant "D", "DG", "DS", "DSG", respectively.

For intra-annual variations, Fig. 15 shows the relative changes of annual and seasonal discharge and different runoff components under SSP585 scenario in the far future period compared to the historical period at the Nuxia station estimated by four calibration variants. With regard to different calibration variants, the similar result was that the reduction of snowmelt runoff (the orange column) in the far future period was most remarkable in spring and summer, while the decrease of glacier melt runoff (the green column) was most significant in summer. The Calibration variant "DG" estimated most significant decreasing snowmelt runoff in spring (–63.1mm, -35.8%), and the variant "D" estimated most significant decreasing snowmelt runoff in summer (–71.3mm, 78.1%). The annual decrease in summer glacier melt runoff was most marked in variant "DS" (–75.0mm, -92.0%). Meanwhile, despite the decreasing snowmelt and glacier melt runoff, the discharge in the YTR basin in the far future period was expected to increase in all the four seasons, mainly owing to the increasing rainfall. The rainfall runoff was estimated to increase in the future evidently in spring, summer and autumn, especially in summer (＞270mm, ~25% in all variants).

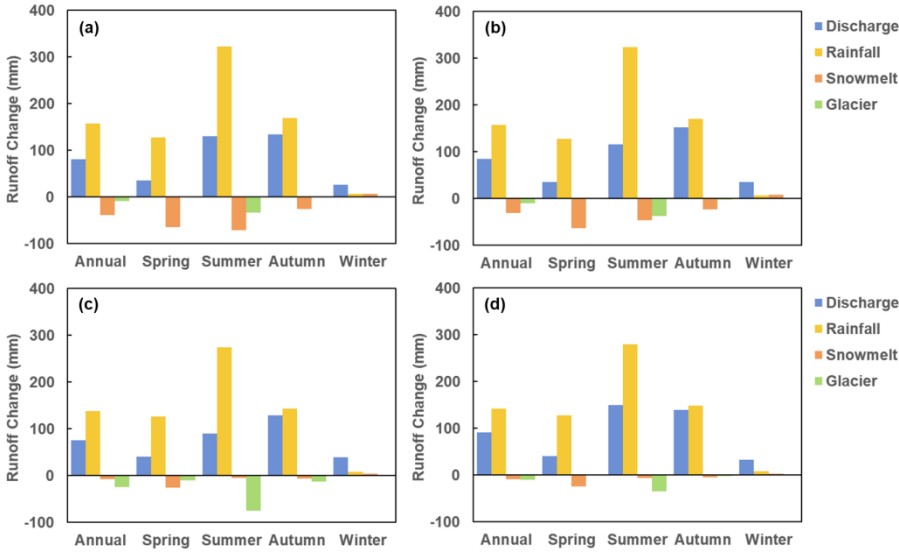

**Figure 15** Relative changes of annual and seasonal discharge and different runoff components under SSP585 scenario in the far future period (2070–2099) compared to the historical period (1980–2009) at the Nuxia station estimated by four calibration variants. (a)–(d) for calibration variant "D", "DG",

"DS", "DSG", respectively.

As for spatial diversity, the changes of different runoff components at upstream stations were further examined. Figure 16 shows the average contributions of different runoff components to discharge in different periods under SSP585 scenario at all stations in the YTR basin estimated by calibration variant "ALL". Similar to the results above at Nuxia station, the contributions of snowmelt and glacier melt runoff at upstream stations all displayed a significant decrease trend under SSP585 scenario in the far future period. Up to the far future, the sum contribution of snowmelt and glacier melt runoff could decrease from ~35% to ~10% at Lazi station, which possessed the highest contribution of melting runoff in the historical period, and from over 10% to less than 5% at other stations (Nugesha, Yangcun and Nuxia) under SSP585 scenario. On the whole, the future variations of runoff and its components at upstream stations were consistent with the outlet station.

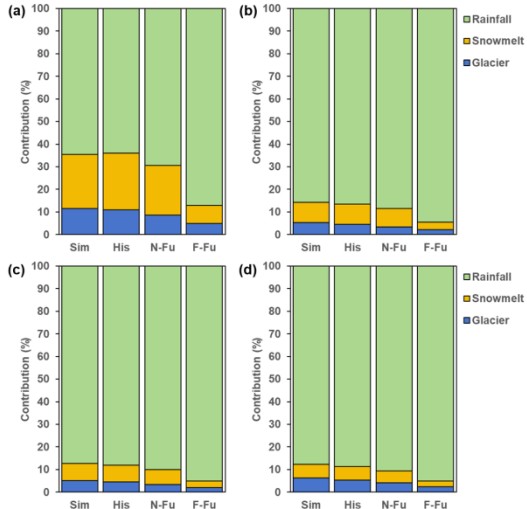

**Figure 16** Contributions of different runoff components to discharge in the calibration period (i.e. 1980–2018, represented by "Sim"), and the historical period (1980–2009), near future period (2020–2049), and far future period (2070–2099) under SSP585 scenario (represented by "His", "N-Fu" and "F-Fu", respectively) at four stations in the YTR basin estimated by calibration variant "ALL". (a)–(d) for Lazi, Nugesha, Yangcun, Nuxia station, respectively.

## 4. Discussion

### 4.1 Influence of runoff component apportionment on streamflow projection

Four different calibration variants for the whole basin were adopted in this study to examine the effects of various observational datasets on the model simulation, and the contributions of different runoff components and the future streamflow projected by the model calibrated under each calibration variant were assessed furthermore. Compared to the variant utilizing all the observational data for calibration, the main differences of other variants could be attributed to two situations: one is the variant with snow unconstrained and the other is the variant with glacier unconstrained. It was observed that in the case of unconstrained snow, the contribution of snowmelt runoff to discharge was relatively high, while in the case of unconstrained glacier, the contribution of glacier melt runoff was relatively high in the historical period, which might be overestimated apparently compared to the actual situation. Furthermore, adding the observational datasets of upstream stations for calibration could further improve the distribution of the model and reduce simulation deviations in different regions within the basin.

For the future projection, the streamflow simulated by models under different calibration variants was similar in general in terms of interannual variation and average seasonal distribution. However, the overestimate of the contribution

of snowmelt and glacier melt runoff could lead to underestimation of the increasing trends of future runoff by approximately 5~10%. The reduction of snowmelt runoff was more marked in the projection under the variant with snow unconstrained and similar results occurred in the projection under the variant with glacier unconstrained, in which the decrease of glacier melt appeared to be more significant.

The calibration variants had an impact on the variation trend of streamflow in the near future period and under low emission scenario (SSP126), while the impact was not significant in the far future period and under high emission scenarios (SSP245 and SSP585). Under all calibration variants, the total streamflow would significantly increase in the far future, along with the overwhelmingly dominated role of rainfall runoff in the streamflow and the substantially reduced meltwater runoff. Furthermore, the significant decrease in snowmelt and glacier melt runoff as well as their contributions to streamflow in the future also occurred to the upstream stations. Altogether, it is beneficial to utilize more observational data to constrain the model in calibration, to obtain better simulation results and understand more accurate contributions of runoff components, so as to obtain more reliable projection of future streamflow's change and changes of various elements.

## 4.2 Comparison with other studies

Table 11 summarizes the contributions of snowmelt and glacier melt runoff to discharge and future projection results in the YTR basin in previous studies and this study. Various hydrological models with different characteristics were used in the hydrological simulation of the YTR basin, including SRM, SPHY, VIC, CREST etc., and divergences existed in the results of runoff component apportionment and future streamflow projection. For instance, the contribution of snowmelt and glacier melt runoff to the total runoff could both range from less than 10% to over 30%. In some studies, the contribution of snowmelt runoff was significantly higher than that of glacier melt (e.g. Zhang et al., 2013 and Su et al., 2016), while some other studies presented the opposite situation with glacier melt runoff taking a larger contribution than the snowmelt (e.g. Lutz et al., 2014 and Feng, 2020). Nevertheless, the contributions of snowmelt runoff and glacier melt runoff were close in some studies (Chen et al., 2017), and some others did not distinguish between the two components or only considered one of them (e.g., Bookhagen and Burbank, 2010 and Gao et al., 2019). Moreover, some of the previous studies also carried out the future runoff's projection in the YTR basin, most of which used the CMIP5 GCMs, while the results of future streamflow changes, including the changes of snowmelt and glacier melt runoff also differed.

In comparison, the contributions of snowmelt and glacier melt runoff to the total runoff in the YTR Basin in our study, constrained by all observational data (discharge, SWE, SCA and GMB), are lower than the results in most previous studies. The divergence of the results could be attributed to several factors. The first and most critical factor is the data used to force and calibrate the model. Constraining the model parameters by the observation datasets related to snow and glacier could brought confidence to the runoff component partitioning. Our results indicated that calibrating the model without snow depth and glacier mass balance datasets resulted in overestimation of meltwater, which was consistent with the fact that the studies not adopting these two datasets estimated much higher contribution of meltwater than our study (e.g., Zhang et al., 2013).

The second factor is the definition of runoff component. Although the terms "snowmelt runoff" and "glacier melt runoff" were adopted in all the studies, they actually referred to different things. Our study considered snow and glacier meltwater as input water sources, while the baseflow from groundwater was not considered as a component. This is because the groundwater was fed by the infiltrated water, which could be finally tracked to the three water sources. But some studies regarded the baseflow as a coordinate component with rainfall and meltwater (e.g., Lutz et al., 2014), thus the rainfall/meltwater runoff in those studies may only refer to the surface runoff induced by the corresponding water source. The results also depended on the calculation equation of the reported contribution ratio. For example, Chen et al. (2017) adopted the similar definition as us and utilized SCA, SWE and total water storage datasets to constrain snow and glacier

simulation, but the contribution ratio was about twice of our results. This is because they calculated the contribution by dividing the meltwater by the total streamflow, which was about half of the denominator adopted in our study (the sum of rainfall, snowmelt and glacier melt) due to evaporation.

Furthermore, the simulation of snow and glacier processes also influenced the runoff component. For instance, if the sublimation during snowfall was not simulated, the contribution of snowmelt runoff may be overestimated. Also, whether to consider the glacier area and how to simulate its changes could also impacted the results (e.g. Immerzeel et al., 2010, Lutz et al., 2014, Gao et al., 2019). If the influence of reduction in glacier area exceeds that of the acceleration of glacier melting caused by rising temperature, the amount of glacier melt runoff would decrease, then affecting the total runoff variation (e.g. Immerzeel et al., 2010 and this study). On the contrary, the situation that the reduction of glacier area was offset by the acceleration of glacier melting might lead to different results of the streamflow change (e.g. Lutz et al., 2014).

As for the future projection, in addition to the differences discussed above, the factors affecting the model results also included the differences between CMIP5 and CMIP6 data, whether the GCMs data was corrected and the reference for correction, as well as the chosen projection period. For example, the precipitation was overestimated for WATCH forcing data (WFD) in the TP, and using it as for GCMs data's correction would lead to a higher streamflow in the future (e.g. Xu et al., 2019). And the changes of streamflow had different variations in different time periods, as our study presented. Generally, the streamflow exhibited an increase trend in the far future, but in the near future, the variation might be different (e.g. Immerzeel et al., 2010, Su et al., 2016, Zhao et al., 2019).

The results were also compared with the studies in other mountainous regions across the world. The streamflow was commonly projected to increase significantly in mountainous river basins, but the mechanism for the increasing trend could be different. In the YTR basin where rainfall dominated the runoff, the projected runoff was mainly determined by the trend of precipitation. On the contrary, in the basins where meltwater contributed significantly to streamflow, the runoff trend was more related to that of temperature, and the runoff might increase event if the precipitation decreased (Slosson et al., 2021). The contribution of meltwater could be especially significant in regions where precipitation and heat are asynchronous, such as Pamir Mountains and Pan-Arctic regions (Pohl et al., 2015; Zhang et al., 2023).

**Table 11** Contributions of snowmelt and glacier melt runoff to discharge and future projection results modelled in the YTR basin in previous studies

| Relevant studies /Reference | Hydrological Model | Data for calibration, hydrological station used | Period | Streamflow contribution | Future projection, future streamflow changes[a] | Notes |
|---|---|---|---|---|---|---|
| Bookhagen and Burbank, 2010 | SRM, based on satellite-derived snow cover, surface temperature, and solar radiation | Observed discharge; not mentioned the calibration station | 2000–2007 | Snow and glacier melt (without distinction): 34.3% (May–Oct: 29.1%) | No | Discharge = rain + snow − ET |
| Immerzeel et al., 2010 | SRM | Observed discharge; not mentioned the calibration station | 2000–2007 | Snow and glacier melt (without distinction): 27% | Yes, use 5 GCMs (A1B scenario, 2046–2065) Streamflow ↓ (19.6%, the best-guess glacier scenario) Rainfall ↑ Glacier ↓ | |
| Zhang et al., 2013 | VIC-glacier (VIC combined with a degree-day glacier algorithm) | Observed discharge; Nuxia | 1961–1999 | Snow: 23.0% Glacier: 11.6% | No | |
| Lutz et al., 2014 | SPHY, with a degree-day snow and glacier melting model | Observed discharge; not mentioned the calibration station | 1998–2007 | Snow: 9.0% Glacier: 15.9% (Rainfall: 58.9% Baseflow: 16.2%) | Yes, use 4 CMIP5 GCMs (RCP4.5/8.5, 2041–2050) Streamflow ↑ (4.5/5.2%) Snow: 7.8/7.2% ( ↓ ) Glacier: 13.7/13.6% ( ↓ ) Rainfall: 61.4/61.6%( ↑ ) Baseflow: 17.5/17.6% ( ↑ ) | Runoff = rainfall + snow melt + glacier melt + baseflow |

| Su et al., 2016 | VIC-glacier | Observed discharge and precipitation; Nuxia | 1971–2000 | Snow: ~23% Glacier: ~12% | Yes, use 20 CMIP5 GCMs (RCP2.6/4.5/8.5, 2011–2040, 2041–2070) Streamflow ↑ Rainfall ↑ Snow ↓ Glacier ↑ Contribution of snow and glacier melt: total−, Snow ↓ Glacier ↑ | |
|---|---|---|---|---|---|---|
| Chen et al., 2017 | CREST (improved) | Observed discharge, SWE, SCA, satellite-derived TWS (total water storage); Nuxia | 2003–2014 | Snow: 10.6% Glacier: 9.9% | No | Total runoff = rainfall + snow meltwater + glacier meltwater − outflow of held water |
| Gao et al., 2019 | HBV | Observed discharge; Nuxia | 1971–2000 | Snowmelt-induced runoff: 24.1~31.4% | Yes, use 18 CMIP5 GCMs (RCP2.6/8.5, 2041–2070, baseline period: 1971–2000) Snowmelt-induced runoff ↓ (8.6/13.1%) | Total runoff = Rainfall − induced runoff + Snowmelt − induced runoff |
| Zhao et al., 2019 | VIC-CAS (coupled with glacier melting and glacier response schemes) | Observed discharge, Glacier distribution; Nuxia | 1971–2010 | Snow: 23.1% Glacier: 5.5% | Yes, use 5 CMIP5 GCMs (RCP2.6/8.5, 2011–2100) Streamflow ↑ Rainfall ↑ Snow ↓ Glacier ↓ | |
| Xu et al., 2019 | THREW | Observed discharge; Nuxia, Bahadurabad | 1980–2001 | Snow: 20.3% Glacier: 5.3% | Yes, use 5 RCMs (RCP4.5/8.5, 2020–2035) Streamflow ↓ (4.1%) / ↑ (19.9%) Snow: 24.6/20.3% Glacier: 6.1/5.0% Rainfall: 69.3/74.8% | WATCH forcing data for bias-correction Runoff = rainfall + Snowmelt + glacier − Evaporation |
| Tian et al., 2020 | THREW | Observed discharge, SWE; Nuxia | 2001–2015 | Snow: 20.0% Glacier: 14.0% | No | |
| Wang et al., 2021 | VIC-glacier | Observed discharge, PET; Nuxia | 1984–2015 | Snow: 15% Glacier: 14% | No | Considering the process of wind blowing snow |
| Cui et al., 2023 | THREW (modified) | Observed discharge, SCA, GMB, Glacier coverage; Nuxia | 1985–2014 | Snow: 12.7% Glacier: 4.4% | Yes, use 22 CMIP6 GCMs (warming levels of 1.5/2.0/3.0°C) Streamflow ↑ Rainfall ↑ Snow ↓ Glacier ↓ / ↑ Contribution: Rainfall ↑ Snow ↓ Glacier− | |
| Guo, 2021 (Master's thesis) | SWAT | Observed discharge, SWE, SCA; Lazi, Nugesha, Yangcun, Nuxia | 2001–2014 | Snow:21.96/6.53/1.91 /4.11% and all ↑ (for the four sub-regions divided by stations) | No | Taking the snow sublimation into account |
| Xuan, 2019 (Doctoral thesis) | SWAT | Observed discharge; Nugesha, Yangcun, Nuxia | 1979–2008 | Snow: 20/20/38% Rainfall: 44/47/32% Groundwater: 36/33/30% (for Nugesha 1974 / Yangcun 1961 / Nuxia 1961) | Yes, use 5 GCMs (RCP2.6/8.5) Rainfall ↑ Snow ↑ / ↑ / ↓ Groundwater ↓ | Runoff = groundwater + rainfall − induced-runoff + snowmelt − induced runoff |
| Wang, 2019 (Doctoral thesis) | GBEHM | Observed discharge, Thickness of frozen ground; Nuxia | 1981–2010 | Glacier: ~5% and ↑ | Yes, use 5 CMIP5 GCMs (RCP4.5, 2011–2060) Streamflow ↑ Rainfall ↑ Evaporation ↑ | Focus on frozen ground degradation |
| Feng, 2020 (Doctoral thesis) | SPHY | Observed discharge; Nuxia | 1980–2014 | Snow: 7.8% Glacier: 30.8% (Rainfall: 52.4% Baseflow: 9.3%) | No | Runoff = rainfall + snow melt + glacier melt + baseflow |
| This study | THREW | Observed discharge, SWE, SCA, GMB; Lazi, Nugesha, Yangcun, Nuxia | 1980–2018 | Snow: 23.9/8.9/7.5/6.0%, Glacier: 11.6/5.3/5.1/6.2% (for the drainage areas of 4 stations) | Yes, use 10 CMIP6 GCMs(SSP126/245/585, 2020–2049/2070–2099) Streamflow ↓ / ↑ Rainfall ↑ Snow ↓ Glacier ↓ | Runoff = rainfall + Snowmelt + glacier − Evaporation |

| | | | | (under the "ALL" calibration variant) | Contribution: Rainfall ↑ Snow ↓ Glacier ↓ | |
|---|---|---|---|---|---|---|

a: The notations " ↑ " / " ↓ " / "−" represent showing a trend of increasing / decreasing / generally unchanged.

## 4.3 Limitations and perspectives

This study constructed the distributed hydrological model THREW in the YTR basin, and set various calibration
variants to compare the constraint effects of different datasets on the model and analyze the streamflow components and
future runoff changes estimated under different variants. However, there are still some limitations in the current research,
which can be further improved in subsequent studies. For instance, the current model reproduced the snow and glacier
melting processes well, and newly considered the sublimation of snowfall, with abundant datasets (observed discharge,
SWE, SCA, and GMB) to calibrate it. But the calculation of snow sublimation as well as the conversion of snow depth
data to SWE were referred to previous study, and the calculation might be a bit rough. Meanwhile, more processes and
corresponding data could be incorporated into the hydrological processes, such as the contribution of frozen soil.
Secondly, our model calibration procedure focused more on the total streamflow and the overall performance on all
objectives, paying less attention to the simulations on extreme events and peak flow processes. The model produced a
generally underestimated peak flow, even in the variant "D" where the NSE for streamflow was higher than 0.9. Results
are similar in some other hydrological modeling studies in the major river basins on the TP (e.g., Su et al., 2023; Xu et al.,
2019). Such simulation bias could be due to either the limitation of daily-scale modeling, or the uncertainties in
precipitation dataset. In specify, the mainstream precipitation datasets generally underestimated the precipitation amount
on the TP, especially the extreme events, because of the lack of validation toward observation data in high altitude regions
where precipitation amount was generally high (Xu et al., 2017; Lyu et al., 2024). Higher resolution simulation and more
accurate forcing datasets would be helpful for improving the simulation of extreme peak events.
Furthermore, different GCMs showed significant divergence in terms of future precipitation and temperature even
after bias correction, leading to large uncertainty ranges in the projected streamflow (Figures 9 and 10). For now, the
ensemble average value of the simulated streamflow forced by different GCMs was regarded as the projection result.
Although this was a commonly used method in similar studies (e.g., Cui et al., 2023), the conclusion was highly dependent
on the quality of the selected GCMs. Improvements in general circulation models and a more comprehensive understanding
of the bias characteristics of GCMs would have been helpful for better streamflow projections.
Last, our discussion in this study mostly focused on the annual discharge at the outlet station of the YTR basin.
Although some seasonal characteristics and results at upstream stations were also mentioned, the analysis of them was
relatively limited. On a more detailed time and spatial scale, there would be more complex variations in runoff changes
and its components. So, the subsequent studies could further analyze the runoff changes and its components in different
regions within the basin, as well as their characteristics on a smaller time scale. Moreover, the current study mainly focused
on the runoff changes and did not consider more socio-economic factors. Yet if combining more factors for analysis, like
the population distribution and water demand situation, more practical conclusions may be obtained.

## 5. Conclusion

The distributed hydrological model THREW was constructed in the YTR basin to analyze the runoff components and
estimate future runoff changes. Different calibration variants were set up to compare the constraint effects of each dataset
and their impacts on the results. The main findings are as follows:
1.  In historical periods, there was no significant changes in annual runoff in the YTR basin over the past six decades,
with a decrease in upstream stations and an increase in the outlet station. The THREW model constrained by

streamflow, snow and glacier datasets indicated that the contributions of snowmelt and glacier melt runoff to streamflow were relatively low for the whole basin, both accounting for about 5~6%. Concretely, the contributions of snowmelt/glacier melt runoff to streamflow were 23.9/11.6%, 8.9/5.3%, 7.5/5.1%, and 6.0/6.2% for Lazi, Nugesha, Yangcun and Nuxia station, respectively.

2. In the future periods, the annual runoff in the YTR basin exhibited an increase trend, not significantly under the low emission scenarios (SSP126 and SSP245) while significantly under the high emission scenario (SSP585), which occurred at all stations. The relative change of annual streamflow depth in the far future period (2070–2099) compared to the historical period (1980–2009) was 26.6mm (102.8%), 50.3mm (57.7%), 76.2mm (51.0%) and 94.6mm (39.9%) at Lazi, Nugesha, Yangcun and Nuxia station, respectively under the high emission scenario. Furthermore, the amounts and contributions of snowmelt and glacier melt runoff would decrease markedly, with their combined contribution reaching less than 10% at Lazi station and less than 5% at other stations in the far future under the high emission scenario.

3. Comparing the results of different calibration variants, it was suggested that using more data to calibrate the model played a vital role in reducing the uncertainty of hydrological simulation. The simulation of SWE, SCA, and GMB all could exhibit a significant bias due to the lack of corresponding observational data to constrain the modeling, resulting in the overestimated contributions of snowmelt and glacier melt runoff to streamflow, for nearly 17% and 10%, respectively at the outlet station. Moreover, the overestimation on the contribution of meltwater runoff led to an underestimation of the increasing trends of annual runoff by approximately 5~10% in future projection, along with a faster reduction of the meltwater runoff.

This study provides a relatively reliable reference for streamflow changes and runoff components during both historical and future periods in the YTR basin, owing to the use of multiple datasets to constrain simulation uncertainties. In the future, the study could potentially be further improved through the incorporation of a more physically based cryospheric module, more accurate input data, and a more comprehensive analysis of streamflow change patterns.

**Acknowledgments**

This study has been supported by the National Natural Science Foundation of China (grant nos. 52309023 and 51825902), the China Postdoctoral Science Foundation (2024T170488) and the Shuimu Tsinghua Scholar Program.

**Code and data availability**

The CMIP6 model outputs are available at https://esgf-node.llnl.gov/search/cmip6/. The ERA5-Land data is available at https://cds.climate.copernicus.eu/cdsapp#!/dataset/reanalysis-era5-land?tab=overview (Muñoz Sabater, 2019). Other datasets for this study are publicly available as follows: CMFD (https://doi.org/10.11888/AtmosphericPhysics.tpe.249369.file, Yang et al., 2019), glacier inventory (https://doi.org/10.3972/glacier.001.2013.db, Liu, 2012), glacier elevation change (https://doi.org/10.6096/13, Hugonnet et al., 2021), snow depth (https://doi.org/10.11888/Snow.tpdc.271743, Yan et al, 2021), snow cover (https://doi.org/10.1016/j.rse.2018.06.021, Chen et al., 2018), LAI (https://lpdaac.usgs.gov/products/mod15a2hv006/, Myneni et al., 2015), NDVI (https://doi.org/10.5067/MODIS/MOD13A3.006, Didan, 2015), and soil property (https://doi.org/10.1029/2019ms001784, Dai et al., 2019). The simulated streamflow, snow water equivalent, snow cover, and glacier mass balance data produced by the model will be available at the Zenodo website at the time of publication.

**Author contributions.** MJZ and YN conceived the idea and collected the data. MJZ, YN and FT conducted the analysis and wrote the paper.

*Competing interests.* At least one of the (co-)authors is a member of the editorial board of Hydrology and Earth System Sciences.

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
