# Peer review of "Runoff component quantification and future streamflow projection"

_EGUsphere, 2024_

## Author Comment (AC1)

**Response to anonymous reviewer #1**

**Overall comment:**

I appreciate the opportunity to review the manuscript, entitled "Runoff component quantification and future streamflow projection in a large mountainous basin based on a multidata-constrained cryospheric-hydrological model." The topic of this study is of great importance not only to the earth and environmental science community but also to policymakers and practitioners such as hydropower companies and water resource managers. This study presents an attempt to systematically analyze streamflow variations and runoff components in the Yarlung Tsangpo River (YTR) basin using a physically-based hydrological model validated by streamflow and multiple datasets related to cryospheric processes. Despite some limitations, the proposed method is capable of reconstructing the sediment yield over the past decades with satisfactory performance.

Overall, I like the study and would recommend a moderate revision before publication. Below are my major and specific comments.

**Response:**

Thank you very much for your high evaluation on our manuscript and the constructive suggestions. We will revise the manuscript thoroughly according to your comments.

**Major Comments**

**Comment 1: Model Validation and Result Presentation**

Based on the modeling scheme, model validation should target multiple hydrological processes, including streamflow, snow, and glaciers. The authors used multiple datasets to validate the model, which is commendable. However, in the results section, the authors seem to focus primarily on streamflow validation. It is suggested that the authors provide a more detailed presentation of the validation results for snow water equivalent (SWE), snow cover area (SCA), and glacier mass balance (GMB) to comprehensively evaluate the model's performance.

**Response:**

Thanks for your comments. Actually, when presenting the results obtained by four calibration variants with different calibration objectives (i.e., D, DS, DG, and DSG), the simulated SWE, SCA and GMB are presented. But we didn't show these results of upstream regions produced by the DSG calibration variant when comparing DSG and ALL variants, only showing the evaluation metrics. We will add more details of simulated SWE, SCA and GMB in the revised manuscript.

**Comment 2: Simulation of Extreme Events**

Typically, extreme hydrological events contribute significantly to annual runoff and can have severe socio-economic impacts. However, daily-scale models often underestimate these extreme events. Have the authors considered the model's performance in simulating extreme events? If there is underestimation, can the model structure or parameterization be improved to enhance simulation accuracy?

**Response:**

Thanks for your comments. Indeed, our study focused on the overall model performance on hydrograph simulation, paying less attention to the extreme events. Besides, based on the comparison of observed and simulated hydrograph, we can observe a generally underestimated peak flow, even in the D calibration variant where NSE is higher than 0.9. Results are similar in some

hydrological modelling studies in other basins on the Tibetan Plateau (e.g., Su et al., 2023, Xu et al., 2019). In addition to the limitation of daily-scale modelling, this could be also due to the uncertainties in precipitation dataset. The mainstream precipitation datasets generally underestimate the precipitation amount, due to the lack of validation toward observation data in high altitude regions, where precipitation amount is generally high (Xu et al., 2017). We will add this issue in the limitation section in the revised manuscript.

**Comment 3: Uncertainty Analysis in Future Projections**

The authors used data from 10 CMIP6 global climate models for future projections, which is a robust approach. However, significant differences may exist between different climate models. It is recommended that the authors provide a more detailed analysis of these inter-model differences and discuss how they affect the uncertainty in future streamflow projections.

**Response:**

Thanks for your comments. For now we only show the uncertainty bands produced by different GCMs in the figure of streamflow projection results. We will describe the uncertainties in the GCMs and the projected future streamflow with more details in the revised manuscript.

**Comment 4: Focusing on the different time periods**

The authors adopt different time periods in different parts, which makes me really confused. For example, 1980-2018 is used for model calibration and validation, 1960-2020 is used for the historical trend analysis, 1960-2014 is the historical period of CMIP6 (L202), and 1980-2009 was used as the baseline of historical simulation. I suggest authors to summarize the different time period adopted in different analyses, and clarify the reason why different time periods are adopted.

**Response:**

Thanks for your comments. Different time periods were adopted in different part due to the availability of streamflow and meteorological datasets. In specify, we selected the past 6 decades (1960-2020) to analyze historical streamflow changes based on the start time of the measurement at hydrological stations, while the most applicative precipitation data over the YTR basin to build the model covered 1979-2018, which was divided into calibration/validation periods (1980-2009/2010-2018) in our study. The CMIP6 data was divided into historical and future periods by 2014 and we chose 2 periods in the near/far future (2020-2049/2070-2099) to compare with the historical 30 years (1980-2009). This indeed makes the analysis confusing. We will add several sentences in the revised manuscript to clearly clarity the adoption of different time periods and the reason for such difference.

**Specific Comments**

Abstract: The abstract can be substantially shortened to one paragraph. It should focus on the study's innovation and main findings.

**Response:**

Thanks for your suggestion. We will shorten the abstract to focus on the innovation and main findings in the revised manuscript.

Introduction: The introduction lacks acknowledgment of existing literature on multi-decadal sediment observations in other high mountain areas and cold regions such as the Andes and the Arctic. Supplementing this literature would enhance the comprehensiveness of the research

background.

**Response:**

Thanks for your suggestion. We will add some references related to water and sediment processes in other could mountainous regions (e.g., Slosson et al., 2021; Zhang et al., 2023) to enhance the paper's relevance to a wider audience.

Methods: The methods section should detail how multiple datasets were integrated into model calibration and how the weights of each dataset were determined.

**Response:**

Thanks for your suggestion. Actually, the datasets for calibration in Table2 was used separately to calculate the evaluation indicators with the model outputs in model calibration. For instance, we used the observational and the simulated streamflow data series to calculate NSE and lnNSE indicators. Then, when selecting the optimal parameters, these calculated indicators owned the same weight, while also taking into account manual judgment. We will explain it more clearly in the revised manuscript.

Discussion: The discussion section could delve deeper into the mechanisms by which climate change affects runoff components, particularly the reasons for the reduction in contributions from snowmelt and glacier melt. Additionally, comparing the findings with studies from other cold regions would enhance the depth of the discussion.

**Response:**

Thanks for your suggestion. The reason for the reduction in meltwater contributions was explained in the result section ("The decreasing snowmelt runoff was due to the reduced snowfall caused by climate warming, while the reduced glacier melt runoff indicated that the effect of shrinking glacier areas was more dominant than the acceleration of glacier melting caused by global warming."), and the discussion section mainly focus on the influence of runoff component estimation on future runoff projection.

It is a good idea to compare the findings with studies with other cold regions. We find that the streamflow is commonly projected to increase in mountainous river basins across the world (e.g., Slosson et al., 2021; Zhang et al., 2023), but the reason for the increasing trend could be different. In the YTR basin where rainfall dominates the runoff, the projected runoff is mainly determined by the trend of precipitation. On the contrary, in the basins where meltwater plays important role in runoff generation, the runoff trend is more related with that of temperature, and the runoff might increase even if the precipitation decreases (Slosson et al., 2021). The contribution of meltwater could be especially significant in the regions where precipitation and heat are asynchronous, such as Pamir Mountains and Pan-Arctic regions (Pohl et al., 2015; Zhang et al., 2023). We will add these discussions in the revised manuscript.

Conclusions: The conclusion should more clearly summarize the main findings and point out the study's limitations and future research directions.

**Response:**

Thanks for your suggestion. We will simplify the conclusion section and point out the limitations and future directions.

Figures and Data: The clarity and informativeness of the figures need improvement. For instance, include the summer discharge trends in Figures 7c-d and ensure consistency with the main text.

**Response:**

Thanks for your suggestion. Improving the informativeness of figures is indeed a good idea, but since Figure 7 is a hydrograph figure and there is no significant trend in summer discharge (as shown in Table 6), including trend lines is not suitable for it. Nonetheless, we will add the trend for the future streamflow projections in Figures 9 and 10.

References: It is recommended to supplement relevant references to place this study in a wider framework of when discussing the impact on hydropower and comparing with other related studies. https://doi.org/10.1038/s41561-022-00953-y; https://doi.org/10.1002/hyp.14633; https://doi.org/10.1016/j.geosus.2024.01.001

**Response:**

Thanks for your suggestion. These references are indeed highly related to our research, and we will add them in the revised manuscript.

L24-25: This statement is rather strong. Maybe better to remove it from the abstract and mention it somewhere in the discussion section.

**Response:** Thanks. We will remove this sentence from the abstract.

L108: There are totally more than 20 GCMs in CMIP6, so how did you select these 10 GCMs?

**Response:** We selected these 10 GCMs based on the stability of these data in our model and the rationality of simulation results. Different GCMs have different starting times and perform variously in specific basins and hydrological models. We tried about 20 models and finally chose these 10 models with more reasonable results and stable performance, and we will add this in the revised manuscript.

L117: Provide the full name of PMV.

**Response:** The full name of PMV is passive microwave, and we will add it in the revised manuscript.

L127: "CGM" should be "GCM".

**Response:** Thanks for correction.

L133-134: Difficult to follow. Maybe provide the specific equation here.

**Response:** We will provide the equation in the revised manuscript.

Table 5: In the notes for DG and DS, "calculated" should be "considered".

**Response:** Thanks for correction.

L188: The historical trend analysis seems to be a separate part from the manuscript. Consider adding some transitional text when describing the methods and results.

**Response:** Thanks for your suggestion. As mentioned in the introduction, this study focuses on the streamflow change in YTR basin during the whole period including both historical and future period, and we aimed to conduct a systematic analysis on the streamflow change and runoff component.

Consequently, the historical trend analysis is the basis for this study. We will add several transitional sentences in the method section in the revised manuscript.

Table 7 and 8: There are two "1980-2009" for discharge NSE; add the unit for the RMSE of SWE and GMB.
**Response:** Thanks for correction. We will correct the period and add the unit of SWE and GMB.

L300-301: Delete this sentence, since "ALL" is the most reliable variant, as mentioned several lines below.
**Response:** We will delete this sentence.

Figure 7 and 8: Consider adding the simulations obtained by "DSG" variant (if this makes the figure too large, maybe add them in Supplementary Materials). We cannot know the specific performance on these elements (e.g., overestimation or underestimation) solely based on the NSE and RMSE.
**Response:** Thanks for the suggestion. We will add the related results in the supplementary materials, since there are already up to 16 figures in the main text.

L319-320: Is this "insignificantly" referring to visual or statistical significance?
**Response:** Both. The P value is <0.01 in all time periods under SSP585 scenario while this is not the case for SSP126/245, and we will consider adding it in the revised manuscript.

Overall, this study addresses an important topic with significant implications for understanding hydrological processes in the Tibetan Plateau. The use of multiple datasets to constrain the model is commendable. However, there are areas where the analysis and presentation could be improved. I hope these comments will be helpful in revising and strengthening the manuscript.
**Response:** Thanks again for your appreciation on our study. We will revise the paper thoroughly according to your suggestions.

---

## Author Comment (AC2)

**Response to anonymous reviewer #2**

**Overall comment:**

This manuscript provides a detailed analysis of runoff components and future streamflow projections in the Yarlung Tsangpo River (YTR) basin using a multi-data-constrained cryospheric-hydrological model. The study successfully integrates multiple observational datasets to validate the model, which enhances the reliability of hydrological simulations in a region with complex cryospheric processes. The findings indicate that snowmelt and glacier melt contribute relatively little to the total streamflow compared to previous studies. Here are some suggestions.

**Response:**

Thank you very much for your high evaluation on our manuscript and the constructive suggestions. We will revise the manuscript thoroughly according to your suggestions.

**Comment 1:**

The introduction of this study is too general. It is recommended to carefully review existing literature on runoff changes and model simulations in the Yarlung Tsangpo River basin and identify the gaps. After reading through the manuscript, I believe the highlight of this paper is the use of multiple datasets and objective functions to calibrate the model and the comparison of runoff and its component changes under different scenarios.

**Response:**

Thanks for your suggestion and pointing out the highlight of our study. Actually, there have been many studies on the runoff changes and model simulations in the YTR basin, but there are still inconsistent results among various studies, which are related to differences in hydrological models, data, and analysis methods used, and the understanding and discussion of such differences are not sufficient. Our research provided results with reduced uncertainty by using multiple datasets to constrain the model and other methods, and conducted an detailed analysis on the reason for the differences in the studies on this topic. We will further review the relevant literature and clarify the highlights of our study based on the gaps now. We will add these in the revised manuscript.

**Comment 2:**

The description of the model section in the manuscript is too brief. Please provide a more detailed explanation of how the model represents glaciers and snow cover, and clearly define the terms snowmelt runoff and glacier runoff.

**Response:**

Thanks for your suggestion. In the THREW model, the degree-day method was used to simulate snow and glacier melting, assuming that snow and glaciers melt at different rates (i.e., different degree-day factors), and relevant parameters including temperature thresholds were calibrated. The terms snowmelt and glacier melt refer to meltwater from snow and glaciers, which enters the catchment and drives runoff generation processes without having undergone evaporation. We will add more introduction of the calculation module of the model and clarify the definition of the terms snowmelt runoff and glacier runoff in the revised manuscript.

**Comment 3:**

In the Data and Methods section, please elaborate on how the future meteorological data were biascorrected.

**Response:**

Thanks for your suggestion. We used the bilinear interpolation method to obtain the GCMs data (from 1960 to 2100) from different spatial resolutions to the same resolution (0.1° grid). Then we used the bias correction method (MBCn algorithm), took the reanalysis meteorological data (CMFD for precipitation and ERA5_Land for temperature) as reference values, and selected 1979-2009 as the correction period and 2010-2018 as the validation period to correct the GCMs data. We will add more introduction about the bias-correction methods for the future meteorological data in the revised manuscript.

**Comment 4:**

Please add a discussion section to explore the impact of uncertainties in the historical and future meteorological data used in this study on the model simulations.

**Response:**

Thanks for your suggestion. Indeed, the historical and future meteorological data can have some impact on the uncertainty of model simulations, but this is not the main focus of our study. For now we show the uncertainty bands produced by different GCMs in the figure of streamflow projection results and we will consider adding more discussion about it in revised manuscript.

---

## Author Response (AR1)

**Response to anonymous reviewer #1**

**Overall comment:**

I appreciate the opportunity to review the manuscript, entitled "Runoff component quantification and future streamflow projection in a large mountainous basin based on a multidata-constrained cryospheric-hydrological model." The topic of this study is of great importance not only to the earth and environmental science community but also to policymakers and practitioners such as hydropower companies and water resource managers. This study presents an attempt to systematically analyze streamflow variations and runoff components in the Yarlung Tsangpo River (YTR) basin using a physically-based hydrological model validated by streamflow and multiple datasets related to cryospheric processes. Despite some limitations, the proposed method is capable of reconstructing the sediment yield over the past decades with satisfactory performance.

Overall, I like the study and would recommend a moderate revision before publication. Below are my major and specific comments.

**Response:**

Thank you very much for your high evaluation on our manuscript and the constructive suggestions. We have revised the manuscript thoroughly according to your comments.

**Major Comments**

**Comment 1: Model Validation and Result Presentation**

Based on the modeling scheme, model validation should target multiple hydrological processes, including streamflow, snow, and glaciers. The authors used multiple datasets to validate the model, which is commendable. However, in the results section, the authors seem to focus primarily on streamflow validation. It is suggested that the authors provide a more detailed presentation of the validation results for snow water equivalent (SWE), snow cover area (SCA), and glacier mass balance (GMB) to comprehensively evaluate the model's performance.

**Response:**

Thanks for your comments. Actually, the simulations on SWE, SCA and GMB have been presented in Figures 6 and 8, and the performances have been described in L261-271. However, for the simulations in upstream regions, we only showed the graphs under the variant "ALL", without showing that produced by the variant "DSG". We have added the figures in the Supplementary and add several sentences to describe the results: "*For comparison, the simulations at upstream stations under the variant "DSG" are shown in Supplementary Figures 2 and 3. The variant "DSG" produced an abnormal fluctuation in discharge during baseflow period at upstream stations, resulting in extremely low values of lnNSE. The snow and glacier simulations were also worse than the variant "ALL", showing larger RMSEs for SWE, SCA and GMB simulations.*"

**Comment 2: Simulation of Extreme Events**

Typically, extreme hydrological events contribute significantly to annual runoff and can have severe socio-economic impacts. However, daily-scale models often underestimate these extreme events. Have the authors considered the model's performance in simulating extreme events? If there is underestimation, can the model structure or parameterization be improved to enhance simulation accuracy?

**Response:**

Thanks for your comments. Indeed, our study focused on the overall model performance on hydrograph simulation, paying less attention to the extreme events. Besides, based on the comparison of observed and simulated hydrograph, we can observe a generally underestimated peak flow, even in the D calibration variant where NSE is higher than 0.9. Results are similar in some hydrological modelling studies in other basins on the Tibetan Plateau (e.g., Su et al., 2023, Xu et al., 2019). In addition to the limitation of daily-scale modelling, this could be also due to the uncertainties in precipitation dataset. The mainstream precipitation datasets generally underestimate the precipitation amount, due to the lack of validation toward observation data in high altitude regions, where precipitation amount is generally high (Xu et al., 2017). We have added this issue in the limitation section in the revised manuscript: "*Secondly, our model calibration procedure focused more on the total streamflow and the overall performance on all objectives, paying less attention to the simulations on extreme events and peak flow processes. The model produced a generally underestimated peak flow, even in the variant "D" where the NSE for streamflow was higher than 0.9. Results are similar in some other hydrological modeling studies in the major river basins on the TP (e.g., Su et al., 2023; Xu et al., 2019). Such simulation bias could be due to either the limitation of daily-scale modeling, or the uncertainties in precipitation dataset. In specify, the mainstream precipitation datasets generally underestimated the precipitation amount on the TP, especially the extreme events, because of the lack of validation toward observation data in high altitude regions where precipitation amount was generally high (Xu et al., 2017; Lyu et al., 2024). Higher resolution simulation and more accurate forcing datasets would be helpful for improving the simulation of extreme peak events.*"

**Comment 3: Uncertainty Analysis in Future Projections**

The authors used data from 10 CMIP6 global climate models for future projections, which is a robust approach. However, significant differences may exist between different climate models. It is recommended that the authors provide a more detailed analysis of these inter-model differences and discuss how they affect the uncertainty in future streamflow projections.

**Response:**

Thanks for your comments. For now we only show the uncertainty bands produced by different GCMs in the figure of streamflow projection results. We have added the description on the uncertainties in the corrected GCM data and projected future streamflow in the 2.2 and 3.4 sections of the revised manuscript: "*After bias correction, the overestimation on precipitation and temperature by GCMs was corrected, but uncertainties still existed in different GCMs. In specify, different GCMs produced a 14.3 mm/yr and 0.27°C difference in mean annual precipitation and temperature for historical period. For the future period, these differences increased to 68.32/62.78/102.43 mm/yr for precipitation and 1.01/1.01/1.66°C for temperature under SSP1-2.6, SSP2-4.5, and SSP5-8.5 scenarios, respectively.*", "*The streamflow projections generated by the 10 GCMs exhibited substantial variation, ranging from 60% to 160% of the average streamflow, as indicated by the uncertainty bars in Figure 9. To address this variability, we used the average of the 10 GCMs to represent the ensemble projection result.*"

**Comment 4: Focusing on the different time periods**

The authors adopt different time periods in different parts, which makes me really confused. For example, 1980-2018 is used for model calibration and validation, 1960-2020 is used for the

historical trend analysis, 1960-2014 is the historical period of CMIP6 (L202), and 1980-2009 was used as the baseline of historical simulation. I suggest authors to summarize the different time period adopted in different analyses, and clarify the reason why different time periods are adopted.

**Response:**

Thanks for your comments. Different time periods were adopted in different part due to the availability of streamflow and meteorological datasets. In specify, we selected the past 6 decades (1960-2020) to analyze historical streamflow changes based on the start time of the measurement at hydrological stations, while the most applicative precipitation data over the YTR basin to build the model covered 1979-2018, which was divided into calibration/validation periods (1980-2009/2010-2018) in our study. The CMIP6 data was divided into historical and future periods by 2014 and we chose 2 periods in the near/far future (2020-2049/2070-2099) to compare with the historical 30 years (1980-2009). This indeed makes the analysis confusing. We have added several sentences at the end of methodology section in the revised manuscript to clearly clarity the adoption of different time periods and the reason for such difference: "*Different time periods were adopted in different analyses. In summary, the past 6 decades (1960-2020) was selected as the time period of historical streamflow trend analysis, based on the available time period of measurement streamflow data. The simulation period was selected as 1980-2018 because the most applicable precipitation input dataset over the YTR basin (CMFD dataset) only covered this period, and was further divided into two periods by 2009 for model calibration (1980-2009) and validation (2010-2018). The future projection analysis adopted 1960-2014 and 2015-2100 as historical and future periods, because the CMIP6 GCMs divided the historical and future periods by 2014, while the historical period here had several years of overlap with the simulation period. Consequently, three periods were selected to represent the baseline historical period (1980-2009), near future (2020-2049) and far future (2070-2099).*"

**Specific Comments**

Abstract: The abstract can be substantially shortened to one paragraph. It should focus on the study's innovation and main findings.

**Response:**

Thanks for your suggestion. We have removed some sentences not so important to focus on the study's innovation and main findings.

Introduction: The introduction lacks acknowledgment of existing literature on multi-decadal sediment observations in other high mountain areas and cold regions such as the Andes and the Arctic. Supplementing this literature would enhance the comprehensiveness of the research background.

**Response:**

Thanks for your suggestion. We have added a sentence and several references at the beginning of the introduction to address the importance of streamflow and sediment change in mountainous regions around the world: "*Change in streamflow and sediment in cold mountainous regions around the world has drawn great interest from researchers (Slosson et al., 2021; Zhang et al., 2023).*"

Methods: The methods section should detail how multiple datasets were integrated into model calibration and how the weights of each dataset were determined.

**Response:**

Thanks for your suggestion. The datasets for calibration in Table2 was used separately to calculate the evaluation indicators with the model outputs in model calibration. In the pySOT program, the indicators for different objectives owned the same weight. However, since different evaluation indicators have different dimensions, we repeated the pySOT 100 times to obtain adequate parameter samples, and A final parameter set was selected from the 100 calibrated sets manually based on the overall performance on multiple objectives. We have explained it more clearly in the revised manuscript.

Discussion: The discussion section could delve deeper into the mechanisms by which climate change affects runoff components, particularly the reasons for the reduction in contributions from snowmelt and glacier melt. Additionally, comparing the findings with studies from other cold regions would enhance the depth of the discussion.

**Response:**

Thanks for your suggestion. The reason for the reduction in meltwater contributions was explained in the result section ("*The decreasing snowmelt runoff was due to the reduced snowfall caused by climate warming, while the reduced glacier melt runoff indicated that the effect of shrinking glacier areas was more dominant than the acceleration of glacier melting caused by global warming.*"), and the discussion section mainly focus on the influence of runoff component estimation on future runoff projection.

It is a good idea to compare the findings with studies with other cold regions. We find that the streamflow is commonly projected to increase in mountainous river basins across the world (e.g., Slosson et al., 2021; Zhang et al., 2023), but the reason for the increasing trend could be different. In the YTR basin where rainfall dominates the runoff, the projected runoff is mainly determined by the trend of precipitation. On the contrary, in the basins where meltwater plays important role in runoff generation, the runoff trend is more related with that of temperature, and the runoff might increase even if the precipitation decreases (Slosson et al., 2021). The contribution of meltwater could be especially significant in the regions where precipitation and heat are asynchronous, such as Pamir Mountains and Pan-Arctic regions (Pohl et al., 2015; Zhang et al., 2023). We have added these discussions in the 4.2 section of the revised manuscript.

Conclusions: The conclusion should more clearly summarize the main findings and point out the study's limitations and future research directions.

**Response:**

Thanks for your suggestion. We have removed some sentences not so important to simplify the conclusion. Besides, we have pointed out the study's limitations and future directions in the conclusion: "*This study provides a relatively reliable reference for streamflow changes and runoff components during both historical and future periods in the YTR basin, owing to the use of multiple datasets to constrain simulation uncertainties. In the future, the study could potentially be further improved through the incorporation of a more physically based cryospheric module, more accurate input data, and a more comprehensive analysis of streamflow change patterns.*"

Figures and Data: The clarity and informativeness of the figures need improvement. For instance, include the summer discharge trends in Figures 7c-d and ensure consistency with the main text.

**Response:**

Thanks for your suggestion. Improving the informativeness of figures is indeed a good idea, but since Figure 7 is a hydrograph figure and there is no significant trend in summer discharge (as shown in Table 6), including trend lines is not suitable for it. Nonetheless, we have added the trend for the future streamflow projections in Figures 9 and 10.

References: It is recommended to supplement relevant references to place this study in a wider framework of when discussing the impact on hydropower and comparing with other related studies. https://doi.org/10.1038/s41561-022-00953-y; https://doi.org/10.1002/hyp.14633; https://doi.org/10.1016/j.geosus.2024.01.001

**Response:**

Thanks for your suggestion. These references are indeed highly related to our research, and we have added them in the revised manuscript.

L24-25: This statement is rather strong. Maybe better to remove it from the abstract and mention it somewhere in the discussion section.

**Response:** Thanks. We have removed this sentence from the abstract.

L108: There are totally more than 20 GCMs in CMIP6, so how did you select these 10 GCMs?

**Response:** We selected these 10 GCMs based on the stability of these data in our model and the rationality of simulation results. Different GCMs have different starting times and perform variously in specific basins and hydrological models. We tried 22 models and finally chose these 10 models with more reasonable results and stable performance, and we have added this in the revised manuscript: "*We evaluated the performance of 22 CMIP6 GCM products and finally chose 10 GCMs to conduct this study, based on the stability of these data in the hydrological model and the rationality of simulation results.*"

L117: Provide the full name of PMV.

**Response:** The full name of PMV is passive microwave, and we have added it in the revised manuscript.

L127: "CGM" should be "GCM".

**Response:** Thanks for correction.

L133-134: Difficult to follow. Maybe provide the specific equation here.

**Response:** We have provided the equation in the revised manuscript.

Table 5: In the notes for DG and DS, "calculated" should be "considered".

**Response:** Thanks for correction.

L188: The historical trend analysis seems to be a separate part from the manuscript. Consider adding some transitional text when describing the methods and results.

**Response:** Thanks for your suggestion. As mentioned in the introduction, this study focuses on the streamflow change in YTR basin during the whole period including both historical and future period,

and we aimed to conduct a systematic analysis on the streamflow change and runoff component. Consequently, the historical trend analysis is the basis for this study.

Table 7 and 8: There are two "1980-2009" for discharge NSE; add the unit for the RMSE of SWE and GMB.

**Response:** Thanks for correction. We will correct the period and add the unit of SWE and GMB.

L300-301: Delete this sentence, since "ALL" is the most reliable variant, as mentioned several lines below.

**Response:** We have deleted this sentence.

Figure 7 and 8: Consider adding the simulations obtained by "DSG" variant (if this makes the figure too large, maybe add them in Supplementary Materials). We cannot know the specific performance on these elements (e.g., overestimation or underestimation) solely based on the NSE and RMSE.

**Response:** Thanks for the suggestion. We have added the related results in the supplementary materials, since there are already up to 16 figures in the main text.

L319-320: Is this "insignificantly" referring to visual or statistical significance?

**Response:** Both. The P value is <0.01 in all time periods under SSP585 scenario while this is not the case for SSP126/245, and we have added it in the revised manuscript: "*The runoff increased insignificantly under SSP126 and SSP245 scenarios, while the increasing trend under SSP585 scenario was visible, with the P value <0.01 in all time periods under SSP585 scenario*".

Overall, this study addresses an important topic with significant implications for understanding hydrological processes in the Tibetan Plateau. The use of multiple datasets to constrain the model is commendable. However, there are areas where the analysis and presentation could be improved. I hope these comments will be helpful in revising and strengthening the manuscript.

**Response:** Thanks again for your appreciation on our study. We have revised the paper thoroughly according to your suggestions.

**Response to anonymous reviewer #2**

**Overall comment:**

This manuscript provides a detailed analysis of runoff components and future streamflow projections in the Yarlung Tsangpo River (YTR) basin using a multi-data-constrained cryospheric-hydrological model. The study successfully integrates multiple observational datasets to validate the model, which enhances the reliability of hydrological simulations in a region with complex cryospheric processes. The findings indicate that snowmelt and glacier melt contribute relatively little to the total streamflow compared to previous studies. Here are some suggestions.

**Response:**

Thank you very much for your high evaluation on our manuscript and the constructive suggestions. We have revised the manuscript thoroughly according to your suggestions.

**Comment 1:**

The introduction of this study is too general. It is recommended to carefully review existing literature on runoff changes and model simulations in the Yarlung Tsangpo River basin and identify the gaps. After reading through the manuscript, I believe the highlight of this paper is the use of multiple datasets and objective functions to calibrate the model and the comparison of runoff and its component changes under different scenarios.

**Response:**

Thanks for your suggestion and pointing out the highlight of our study. Actually, there have been many studies on the runoff changes and model simulations in the YTR basin, but there are still inconsistent results among various studies, which are related to differences in hydrological models, data, and analysis methods used, and the understanding and discussion of such differences are not sufficient. Our research provided results with reduced uncertainty by using multiple datasets to constrain the model and other methods, and conducted a detailed analysis on the reason for the differences in the studies on this topic (as discussed in the 4.2 section). We have added several sentences to clarify the current gaps of existed studies in the introduction section: "*However, the contribution of runoff components still had a significant uncertainty among different studies, and a consistent conclusion on this issue has yet to be reached. In specify, the estimated contribution of glacier melt to streamflow in the YTR basin ranged from 3.5% (Wang et al., 2021) to 29% (Boral and Sen, 2020). Besides, the reason for such divergence remained unclear, and the influence of runoff component estimation on future streamflow projection was not investigated adequately. A reliable reference value for runoff components obtained by robust modeling method is crucial for water resource management.*"

**Comment 2:**

The description of the model section in the manuscript is too brief. Please provide a more detailed explanation of how the model represents glaciers and snow cover, and clearly define the terms snowmelt runoff and glacier runoff.

**Response:**

Thanks for your suggestion. In the THREW model, the degree-day method was used to simulate snow and glacier melting, and the volume/area of snow and glacier were dynamically updated based on the balance between accumulation and meltwater. We have provided more details on snow and

glacier simulation in the revised manuscript: "*The snow water equivalent in each REW was updated based on the snowfall and snowmelt, and the snow cover area was then determined by the snow cover depletion curve. To represent the change in meteorological factors along the altitudinal profile of glaciers, each REW was further divided into several elevation bands to simulate the evolution of glaciers. For each glacier simulation unit, processes including the snow accumulation and snowmelt over a glacier, the turnover of snow to ice, and the ice melt were considered. The mass balance of each glacier simulation unit equaled the difference between snowfall and the total meltwater.*"

The terms snowmelt and glacier melt refer to meltwater from snow and glaciers, which enters the catchment and drives runoff generation processes without having undergone evaporation. We have clarified the definition of the terms snowmelt runoff and glacier runoff in the revised manuscript: "*There are two definitions to quantify the contributions of runoff components to streamflow in the THREW model. One was based on the individual water sources in the total water input triggering runoff processes, including rainfall, snowmelt, and glacier melt and another was based on pathways of runoff-generation processes, resulting in surface and subsurface runoff (baseflow) (Nan et al. 2022). Here we focused on the first definition and calculated the contributions of different water sources (rainfall, snowmelt, and glacier melt) to the total runoff. More precisely, the terms snowmelt and glacier melt refer to meltwater from snow and glaciers, which enters the catchment and drives runoff generation processes without having undergone evaporation, and the total discharge was equal to the sum of these three components minus evaporation, thereby achieving the water balance in the THREW model.*"

**Comment 3:**

In the Data and Methods section, please elaborate on how the future meteorological data were bias-corrected.

**Response:**

Thanks for your suggestion. We used the bilinear interpolation method to obtain the GCMs data (from 1960 to 2100) from different spatial resolutions to the same resolution (0.1° grid). Then we used the bias correction method (MBCn algorithm), took the reanalysis meteorological data (CMFD for precipitation and ERA5_Land for temperature) as reference values, and selected 1979-2009 as the correction period and 2010-2018 as the validation period to correct the GCMs data. We have added more introduction about the bias-correction methods for the future meteorological data in the revised manuscript: "*We evaluated the performance of 22 CMIP6 GCM products and finally chose 10 GCMs to conduct this study, based on the stability of these data in the hydrological model and the rationality of simulation results. The basic information of these 10 GCMs is shown in Table 3. The CMIP6 data during 1960–2100 (divided into historical and future periods by 2014) were interpolated from various spatial resolutions into the same degree (0.1° grid) through a bilinear interpolation scheme. The biases in the GCMs data were further corrected against the reanalysis meteorological data (CMFD for precipitation and ERA5_Land for temperature, using 1979–2009 as the reference period for correction, and 2010–2018 for validation) based on a multiplicative bias-correction approach (MBCn algorithm, Alex J. Cannon, 2018; Cui et al., 2023). The average precipitation and temperature of the corrected GCMs are presented in Fig. 2. After bias correction, the overestimation on precipitation and temperature by GCMs was corrected, but uncertainties still existed in different GCMs. In specify, different GCMs produced a 14.3 mm/yr and 0.27 ℃ difference*"

*in mean annual precipitation and temperature for historical period. For the future period, these differences increased to 68.32/62.78/102.43 mm/yr for precipitation and 1.01/1.01/1.66 °C for temperature under SSP1-2.6, SSP2-4.5, and SSP5-8.5 scenarios, respectively.*"

**Comment 4:**

Please add a discussion section to explore the impact of uncertainties in the historical and future meteorological data used in this study on the model simulations.

**Response:**

Thanks for your suggestion. We have added two paragraphs in the limitation section to discuss the impact of uncertainties in historical and future meteorological data. The historical data mainly influences the model performance, especially the peak flow processes, while the future data results in a large uncertainty in the projected streamflow. The added paragraphs are as follows:

"*Secondly, our model calibration procedure focused more on the total streamflow and the overall performance on all objectives, paying less attention to the simulations on extreme events and peak flow processes. The model produced a generally underestimated peak flow, even in the variant "D" where the NSE for streamflow was higher than 0.9. Results are similar in some other hydrological modeling studies in the major river basins on the TP (e.g., Su et al., 2023; Xu et al., 2019). Such simulation bias could be due to either the limitation of daily-scale modeling, or the uncertainties in precipitation dataset. In specify, the mainstream precipitation datasets generally underestimated the precipitation amount on the TP, especially the extreme events, because of the lack of validation toward observation data in high altitude regions where precipitation amount was generally high (Xu et al., 2017; Lyu et al., 2024). Higher resolution simulation and more accurate forcing datasets would be helpful for improving the simulation of extreme peak events.*

*Furthermore, different GCMs showed significant divergence in terms of future precipitation and temperature even after bias correction, leading to large uncertainty ranges in the projected streamflow (Figures 9 and 10). For now, the ensemble average value of the simulated streamflow forced by different GCMs was regarded as the projection result. Although this was a commonly used method in similar studies (e.g., Cui et al., 2023), the conclusion was highly dependent on the quality of the selected GCMs. Improvements in general circulation models and a more comprehensive understanding of the bias characteristics of GCMs would have been helpful for better streamflow projections.*"